# EAGER: ENTROPY-AWARE GENERATION FOR ADAPTIVE INFERENCE-TIME SCALING

## ABSTRACT

With the rise of reasoning language models and test-time scaling methods as a paradigm for improving model performance, substantial computation is often required to generate multiple candidate sequences from the same prompt. This enables exploration of different reasoning paths toward the correct solution, however, allocates the same compute budget for each prompt. Grounded on the assumption that different prompts carry different degrees of complexity, and thus different computation needs, we propose EAGER, a training-free generation method that leverages model uncertainty through token-wise entropy distribution to reduce redundant computation and concurrently improve overall performance. EAGER allows branching to multiple reasoning paths only in the presence of high-entropy tokens, and then reallocates the saved compute budget to the instances where exploration of alternative paths is most needed. We find that across multiple open-source models on complex reasoning benchmarks such as AIME 2025, EAGER can reallocate the budget without accessing target labels, achieving the best efficiency-performance trade-off in terms of reasoning length and Pass@k. When target labels are accessible, EAGER generates up to 65% fewer tokens (hence saving compute) and achieves up to 37% improvement in Pass@k compared to the FULL PARALLEL sampling. Our results show that EAGER consistently maximizes the efficiency-performance trade-off by enabling dynamic control over computation expenditure.

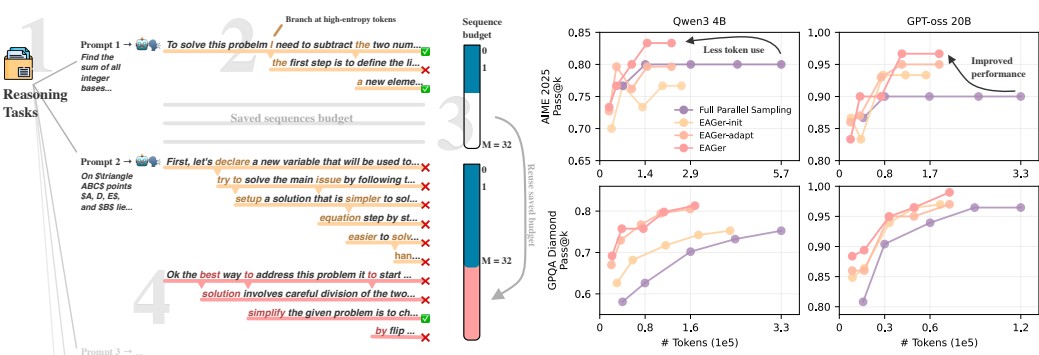

Figure 1: **Left:** We introduce EAGER, a generation method designed for (1) reasoning benchmarks, where decoding adaptively triggers branching at high-entropy tokens (2). Each prompt is assigned a maximum budget of $M$ candidate sequences, and unused budget from sequences that do not saturate their allocation (3) is carried forward. The remaining budget is then reallocated (4) either to prompts that hit the $M$ cap, as in EAGER-adapt, or, when target labels are available, to prompts that have not yet reached a correct solution (Pass@k = 0), yielding our full EAGER. **Right:** Our approaches (EAGER -init, -adapt and full EAGER) consistently reduce token usage compared to the standard FULL PARALLEL sampling approach when scaling the $M$ limit $\in [4, 8, 16, 24, 32]$. In addition, EAGER always achieves a clear performance advantage over all other decoding methods.

# 1 INTRODUCTION

Recent advances in large language models (LLMs) have led to substantial improvements in complex reasoning tasks, particularly with the adoption of chain-of-thought (CoT) prompting (Wei et al., 2022). Such tasks often admit multiple valid reasoning paths that converge to the same correct solution (Stanovich & West, 2002). Rather than relying on a single greedy decoding path, the single generation can be replaced by multiple sampled candidate sequences, thereby producing a diverse set of reasoning paths and corresponding final answers (Wang et al., 2023). This strategy has been shown to enhance performance on challenging reasoning problems: by exploring multiple reasoning paths, the model reduces its reliance on the stochasticity of a single greedy generation and increases the likelihood of arriving at a correct solution.

Despite its success, CoTs introduce an inherent computational inefficiency: reasoning sequences tend to be long, and a large portion of the tokens generated are predictable continuations rather than genuine decision points (Wang et al., 2025). This inefficiency is amplified in approaches that explore multiple reasoning paths in parallel, where each path independently regenerates identical prefixes before diverging. For prompts with simple problems, many of these paths converge to the same solution with little variation, resulting in redundant computation. For more complex prompts, however, the diversity of reasoning paths becomes crucial, and additional generations may be necessary to discover a correct solution (Snell et al., 2025; Muennighoff et al., 2025). This observation suggests that a per-problem decision to let or not let the model explore alternative paths would be desirable. We argue that such decision can be guided by monitoring model uncertainty during generation towards an adaptive allocation of computating budget. Intuitively, when the model's predictions are confident and stable, only a few candidate sequences are needed, while at points of high uncertainty, where multiple reasoning paths are plausible, additional exploration becomes critical.

To address these issues, we introduce **EAGER**, an **Entropy-Aware Generation** method that monitors token-level uncertainty during decoding to guide where new parallel reasoning traces should start. By branching only at high-entropy tokens, we avoid regenerating identical low-entropy continuations, substantially reducing computation overhead without sacrificing coverage of diverse reasoning traces. Furthermore, reducing the parallel samples for *easy* prompts, EAGER dynamically allocates the unused sampling budget towards more challenging ones, maximizing the benefits of inference-time scaling for difficult prompts.[1]

We evaluate EAGER on a diverse set of benchmarks, spanning from complex math problems to science-related questions and code generation tasks. All the tested LMs, from the smallest 3B to the biggest 20B parameter model, show a performance boost of up to 37% when using EAGER compared to our baseline FULL PARALLEL sampling setting.

Our main contributions are as follows:

- We empirically show that token-wise entropy peaks as a form of online (i.e., measured during generation) uncertainty is a good proxy that shows when more exploration is needed during the generation, hence reflecting the difficulty of a prompt for the model used.

- We introduce, *a novel*, training-free decoding method that leverages entropy distribution during generation to dynamically reduce compute cost while maintaining the benefits of inference-time scaling. EAGER generates up to 65% fewer tokens and saves up to 80% of the entire generation budget across all our benchmarks and models.

- To maximize the benefits of inference-time scaling, we show that EAGER enables adaptive use of the given sampling budget, where it spends more compute on the *hard* problems. This yields up to 13% improvements in test-time settings without access to target labels, and up to 37% when labels are available, on the AIME 2025 reasoning benchmark.

# 2 PRELIMINARIES

In the inference-time scaling paradigm, a language model generates multiple parallel sequences so that it can *explore* various reasoning paths to find a correct solution (Welleck et al., 2024; Snell

---

[1]Code and data: released upon acceptance.

et al., 2025). This is oftentimes facilitated by sampling completions from the model with a relatively high temperature using methods such as nucleus sampling (Holtzman et al., 2020). We refer to this standard approach as the FULL PARALLEL sampling generation procedure. This approach is useful in various settings, including for the generation of diverse solutions to a problem, or in large-scale reinforcement learning (RL) pipelines such as in RLVR (DeepSeek-AI, 2025), where among the diverse set of generated sequences, only the correct ones are selected to update the policy.

Our objective is to optimize this process by exploiting uncertainty during generation, allowing an efficient allocation of resources (in terms of number of generated sequences) to solve a given prompt.

## 2.1 UNCERTAINTY IN LLMS' GENERATIONS

Among the different techniques for uncertainty quantification in LLMs, we focus our attention on top-K token entropy, as token entropy has been shown to be a powerful uncertainty quantification measure (Fomicheva et al., 2020). We define top-K token entropy as:

$$H_t^{(K)} := - \sum_{i \in \mathcal{I}_t^{(K)}} p_{t,i}^{(K)} \log p_{t,i}^{(K)}, \tag{1}$$

where $\mathcal{I}_t^{(K)} \subseteq \{1, \ldots, |V|\}$ is the index set of the $K$ tokens with highest $p_{t,i}$ probability with $V$ denoting the vocabulary of the LM and $t \in \mathbb{N}^+$ indexes the current generation step. Specifically, the quantity $p_{t,i}$ represents the probability assigned by the model to token $i \in V$ at step $t$ after the softmax computation. We denote $\mathcal{I}_t^{(K)} \subseteq \{1, \ldots, |V|\}$ the index set of the $K$ tokens with the highest probability $p_{t,i}$ and $p_{t,i}^{(K)}$ their re-normalized probabilities, given by:

$$p_{t,i}^{(K)} := \frac{p_{t,i}}{\sum_{j \in \mathcal{I}_t^{(K)}} p_{t,j}}, \quad i \in \mathcal{I}_t^{(K)}, \tag{2}$$

where $\sum_{i \in \mathcal{I}_t^{(K)}} p_{t,i}^{(K)} = 1$.

Compared to more precise and computationally intensive uncertainty quantification methods found in the literature (Vashurin et al., 2025; Kuhn et al., 2023; Duan et al., 2024, e.g.,), top-$K$ token entropy provides a strong approximation to the entropy of the full-vocabulary, as it computes the dominant contributions from the most probable tokens with minimal computational overhead [2].

## 2.2 ENTROPY IN LONG CHAIN-OF-THOUGHT REASONING

For our goal of saving resources in parallel sampling by leveraging model uncertainty, we first need to determine whether and how token entropy values relate to the model's final performance. To this end, we analyze the entropy patterns of the CoT sequences generated by an LLM to solve challenging problems. We monitor the entropy of each token during generation, and rather than analyzing the entire entropy sequence, we focus on identifying significant spikes as signals of higher uncertainty. We hypothesize that this peak-entropy measure can serve as a proxy for the model's perceived difficulty of a problem and thus its (in)ability to solve it: high peaks indicate moments where the model is highly uncertain about the next step in its reasoning chain, low entropy indicates that the model is more confident about what to generate next.

Given the input prompt $x$, we sample $M$ independent candidate sequences $\{t^{(m)}\}_{m=1}^M$ from the language model. During generation, for each token position $t$ in each sequence $m$, we record the token entropy $H_t^{(K)}(y^{(m)})$, with $K = 20$. For each sequence, we define the peak entropy value $\bar{H}_{\text{peak}}^{(m)}$ as the mean of all entropy values that lie in the $p^{\text{th}}$ percentile of the sequence's entropy distribution:

$$\bar{H}_{\text{peak}}^{(m)}(p^{\text{th}}) := \frac{1}{|\mathcal{T}_m^{\text{peak}}(p^{\text{th}})|} \sum_{t \in \mathcal{T}_m^{\text{peak}}(p^{\text{th}})} H_t^{(K)}(y^{(m)}), \tag{3}$$

---

[2]Full-vocabulary entropy computation can be costly due to large vocabulary sizes, often in the tens of thou-

where

$$\mathcal{T}_m^{\text{peak}}(p^{\text{th}}) := \{t : H_t^{(K)}(y^{(m)}) \geq p^{\text{th}}\left(\{H_{t'}^{(K)}(y^{(m)})\}_{t'}\right)\}, \tag{4}$$

and $p^{\text{th}}(\cdot)$ denotes the $p^{\text{th}}$ percentile of the entropy sequence.

We run Qwen3 4B[3], a strong open-source LLM with long CoT reasoning capabilities, on five standard reasoning benchmarks for math, science and code generation tasks (see Section 4 for the benchmarks' details), allowing for $M = 32$ parallel sequences to be generated.

Figure 2 shows the Pass Rate accuracy, i.e., the proportion of correct answers out of $M = 32$ generations, for each prompt and the corresponding average peak entropy $\bar{H}_{\text{peak}}^{(m)(p^{\text{th}})}$. We focus on the top percentile, specifically $p^{\text{th}} = 99.9$, to isolate the highest entropy peaks. We observe a statistically significant negative correlation ($\rho \approx -0.55$), between the peak entropy during generation and model Pass Rate accuracy. This suggests that higher entropy peaks, indicative of greater uncertainty during generation, are associated with lower performance. Thus, additional path exploration during these phases may help to improve performance. Conversely, when entropy remains low, the model is more confident on the generated solutions (hence, the long CoT reasoning sequences), suggesting that further exploration may be less likely to yield significant improvements. This observation is in line with recent work which found that high-entropy tokens disproportionately contribute to performance gains during RL training (Wang et al., 2025).

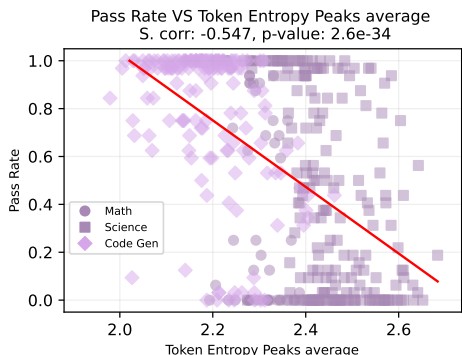

Figure 2: For each sequence generated by Qwen3 4B with FULL PARALLEL sampling ($M = 32$), we report its Pass Rate accuracy and the average entropy peak ($p^{\text{th}} = 99.9$). The results reveal a negative correlation ($r = -0.547$) between Pass Rate and the average entropy peak across sequences. Notably, sequences exhibiting higher entropy at any generation step are less likely to yield a correct answer.

Given this evidence, we ask: *Can token entropy be leveraged to develop a decoding adaptive strategy that allocates more compute to uncertain regions while limiting effort in more confident segments?*

## 3 ENTROPY-AWARE GENERATION EXPLAINED

We introduce **EAGER**, a training-free inference-time scaling approach aimed at optimizing parallel sampling by leveraging token entropy to guide resource allocation. EAGER consists of two stages: in the first one, EAGER-init dynamically adjusts the generation process to focus on sequences where the most effort is needed, while pruning unnecessary generations. In the second stage, the saved computational budget is reallocated to enhance performance on the remaining challenging prompts.

### 3.1 EAGER-INIT: SAVE COMPUTE VIA TOKEN ENTROPY

EAGER-init represents the first stage of our approach and operates by identifying potentially *easy* questions during generation. Instead of sampling constant $M$ generations for every prompt, EAGER-init computes token entropy $H_t^{(K)}$ at each step $t$, and compares it to a predefined threshold $\theta$.[4] If the observed entropy exceeds this threshold, the current sequence is *branched*, creating a new candidate continuation at that position. If the entropy is below the threshold, the generation continues with the existing sequence. During the branching step, we reuse the token distribution from the model but adopt a temporally greedy approach; we select the top two most likely tokens to ensure

---

sands. By restricting calculations to the top-$K$ most probable tokens, the token entropy significantly reduces computational overhead while maintaining efficiency during generation.

[3]https://huggingface.co/Qwen/Qwen3-4B

[4]We empirically find the best threshold for a model. See Section 4.1 for a detailed analysis of the threshold.

that the two new sequences always start with different tokens. This process continues until the total number of active sequences reaches a predefined limit $M$, at which point no further branching occurs. A detailed overview of the EAGER-init algorithm is provided in Algorithm 1.

This process yields a generation tree where the root is the initial sequence, and each branching node corresponds to a high-entropy token (Figure 1); the generation stops when the total number of nodes is equal to $M$. For implementation efficiency, we restrict the branching procedure to long CoT sequences only, and entropy monitoring is halted if no branching has occurred within the previous 1000 tokens from the last branch.

---

**Algorithm 1:** EAGER-init sequence generation

**Input:** Prompt $x$, entropy threshold $\theta > 0$, max active sequences $M$, temperature $\tau$, top-$K$ for entropy $K$, maximum steps $T$
**Output:** Completed set of sequences $\mathcal{Y}$

**Notation:** $H_t^{(K)}$ is the top-$K$ token entropy at step $t$ under distribution $p(\cdot \mid x, y)$.

Initialize active set $\mathcal{A} \leftarrow \{y^{(1)}\}$ ;          // initial continuation from prompt $x$
Initialize completed set $\mathcal{Y} \leftarrow \varnothing$ ;
**for** $t \leftarrow 1$ **to** $T$ **do**
 **if** $\mathcal{A} = \varnothing$ **then**
  | **break**
 **foreach** *sequence* $y \in \mathcal{A}$ **do**
  Compute next-token distribution $p(\cdot \mid x, y)$ with temperature $\tau$ ;
  Compute entropy $H_t^{(K)}$ from top-$K$ probabilities ;
  **if** $H_t^{(K)} \geq \theta$ **and** $|\mathcal{A}| < M$ **then**
   $a_1 \leftarrow \arg\max_a p(a \mid x, y)$ ;                     // most likely token
   $a_2 \leftarrow$ second-most-likely token under $p$ ;
   Update $y \leftarrow y \circ a_1$ ;                     // greedy continuation
   Create branch $y' \leftarrow y \circ a_2$, add $y'$ to $\mathcal{A}$ ;
  **else**
   Sample $a \sim p(\cdot \mid x, y)$ and update $y \leftarrow y \circ a$ ;
  **if** $y$ *ends with EOS or length limit* **then**
   Move $y$ from $\mathcal{A}$ to $\mathcal{Y}$ ;
**return** $\mathcal{Y}$ ;

---

**Reducing test-time compute through EAGER-init.** EAGER-init, saves computational budget through two mechanisms. The first arises directly from the branching logic: if a branch occurs at token position $t$, all preceding tokens $(0, \ldots, t-1)$ are reused across branches rather than being regenerated independently. The second, and more substantial source of savings occurs when the generation process does not saturate the maximum number of sequences $M$ set per prompt. For easy queries, the model's default sampling converges to identical or near-identical completions, so that EAGER-init may terminate with only a single sequence, saving $M-1$ full generations compared to a fixed-budget baseline which would let the model generate $M$ sequences for any given prompt. This surplus capacity can then be reallocated where it is most needed.

## 3.2 EAGER: DYNAMICALLY ALLOCATE THE SAVED COMPUTE

The key challenge is to devise *the best strategy to reallocate the compute which has been saved*. We start by defining an additional budget $b$, computed as:

$$b = \min(M_{\text{theoretical}} - M_{\text{actual}}, 2M) \tag{5}$$

where $M_{\text{theoretical}} = M \times |\mathcal{D}|$ is the maximum possible number of sequences that could be generated for the dataset $\mathcal{D}$, and $M_{\text{actual}} = \sum_{i=1}^{|\mathcal{D}|} \# \text{Seq}_i$ is the total number of sequences actually produced under entropy-aware generation.

---

**Algorithm 2:** Full EAGER algorithm

**Input:** Dataset $\mathcal{D} = \{(x_i, z_i)\}_{i=1}^N$, initial generations $\{\mathcal{Y}_i\}_{i=1}^N$ (from EAGER-init), max sequences per prompt $M$, entropy threshold $\theta$

**Output:** Augmented generations $\{\mathcal{Y}_i'\}_{i=1}^N$

Compute $M_{\text{theoretical}} \leftarrow M \cdot |\mathcal{D}|$ ;

Compute $M_{\text{actual}} \leftarrow \sum_{i=1}^N |\mathcal{Y}_i|$ ;

Set remaining budget $b \leftarrow M_{\text{theoretical}} - M_{\text{actual}}$ ;

Identify challenging prompts $\mathcal{I} = \{i \mid \text{Pass@k}(\mathcal{Y}_i, z_i) = 0\}$ ;

**if** $b = 0$ *or* $\mathcal{I} = \varnothing$ **then**
  |    **return** $\{\mathcal{Y}_i\}$

Assign additional budget $b = \min(b, 2M)$ uniformly across all $i \in \mathcal{I}$ ;

**foreach** $i \in \mathcal{I}$ **do**
  |    **if** $|\mathcal{Y}_i| < M$ **then**
  |   |    // underutilizing prompt
  |   |    Set $\theta' \leftarrow 0.8 \cdot \theta$ ;
  |   |    Generate up to $M + b$ sequences for $x_i$ using Algorithm 1 with $\theta'$ ;
  |    **else**
  |   |    // prompt already saturated at $M$
  |   |    Set $\theta' \leftarrow \theta$ ;
  |   |    Generate up to $M + b$ sequences for $x_i$ using Algorithm 1 with $\theta'$ ;
  |    Append new sequences to $\mathcal{Y}_i$ ;

**return** $\{\mathcal{Y}_i'\}_{i=1}^N$ ;

---

The term $M_{\text{theoretical}} - M_{\text{actual}}$ represents the surplus budget created by early stopping in *easy* prompts. We cap $b$ at $2M$ to avoid pathological cases where extremely large surpluses would lead to disproportionately high generation budgets for single prompts.[5]

We consider two scenarios: (i) a test-time setting where target labels are unavailable and reallocation must rely solely on model signals, and (ii) a setting where target labels are accessible, as in reinforcement learning pipelines where only correct generations are used for policy updates.

**Budget reallocation without target labels.** In the absence of target labels, we identify *saturating prompts*, those that hit the maximum branching cap $M$, as candidates for additional budget. The rationale is that when $M$ serves as a hard limit, promising reasoning paths may remain unexplored. To mitigate this, we start from a low branching threshold ($\theta = 2.0$, see Section 4.1) and reallocate saved budget exclusively to these prompts. We denote this strategy as EAGER-adapt.

**Fine-grained budget reallocation with target labels.** When target labels are available, we instead focus on *challenging prompts*, defined as those that fail to achieve Pass@k accuracy under EAGER-init (i.e., none of the generated sequences match the correct answer).

For *underutilizing prompts* (fewer than $M$ sequences under EAGER-init), we lower the entropy threshold $\theta$ by 20%, encouraging earlier and more frequent branching. For *saturating prompts* (exactly $M$ sequences under EAGER-init), we extend generation up to a new per-prompt limit $M+b$, thereby deepening exploration where additional sequences may yield correct solutions.

Reallocation in this setting is uniform across all failing prompts, while the generation strategy adapts based on each prompt's prior behavior. By redirecting unused capacity from easier prompts to harder ones, this approach increases coverage without exceeding the global budget $M_{\text{theoretical}}$ (see Algorithm 2). Importantly, savings from branch-based token reuse persist even when $b > 0$, and all additional sequences remain governed by Algorithm 1, ensuring that total token usage stays below an equivalent fixed-budget FULL PARALLEL sampling baseline.

---

[5]Especially in larger datasets, budget savings for easy prompts were large enough to allocate hundreds, if not thousand, of additional sequences to single failing prompts; this cap prevents excessive unbalanced allocation.

| Model | Sampling | AIME 2025 | | | GPQA-Diamond | | | HumanEval Plus | | |
|---|---|---|---|---|---|---|---|---|---|---|
| | | p@k | c@k | PR | p@k | c@k | PR | p@k | c@k | PR |
| SmolLM 3B | FULL PARALLEL | 0.53 | 0.00 | 0.06 | 0.49 | 0.00 | 0.03 | 0.00 | 0.00 | 0.00 |
| | EAGER-INIT | 0.53 | 0.07 | 0.11 | 0.59 | 0.10 | 0.15 | 0.68 | 0.46 | 0.44 |
| | EAGER | **0.73** | **0.33** | **0.31** | **0.85** | **0.12** | **0.18** | **0.75** | **0.56** | **0.52** |
| Qwen3 4B | FULL PARALLEL | 0.80 | 0.70 | 0.62 | 0.75 | 0.51 | 0.43 | 0.91 | 0.82 | 0.78 |
| | EAGER-INIT | 0.77 | 0.70 | 0.61 | 0.79 | 0.51 | 0.43 | 0.86 | 0.86 | **0.86** |
| | EAGER | **0.83** | **0.73** | **0.69** | **0.81** | **0.59** | **0.54** | **0.94** | **0.87** | 0.86 |
| DeepSeek 8B | FULL PARALLEL | **0.80** | **0.67** | 0.65 | 0.95 | 0.15 | 0.18 | 0.95 | **0.90** | 0.86 |
| | EAGER-INIT | 0.70 | 0.63 | 0.64 | 0.93 | **0.25** | 0.24 | 0.96 | 0.85 | 0.77 |
| | EAGER | 0.77 | **0.67** | **0.67** | **0.96** | **0.25** | **0.25** | **0.97** | **0.90** | **0.89** |
| GPT-Oss 20B | FULL PARALLEL | 0.90 | **0.83** | 0.67 | 0.96 | 0.68 | 0.65 | 0.95 | 0.83 | 0.79 |
| | EAGER-INIT | 0.93 | 0.80 | 0.66 | 0.97 | 0.71 | **0.66** | 0.97 | 0.88 | **0.85** |
| | EAGER | **0.97** | 0.80 | **0.68** | **0.99** | **0.72** | 0.66 | 0.97 | **0.89** | **0.85** |

Table 1: Comparison of FULL PARALLEL, EAGER-INIT and EAGER in AIME-2025, GPQA-Diamond and HumanEval Plus. We report pass@k, cons@k and Pass Rate where k is number of samples generated (while always 32 for the baseline, differs per prompt for EAGER-init and EAGER). EAGER consistently achieves the best results and EAGER-init performs very competitive with FULL PARALLEL sampling while saving significant amount of compute as shown in Figure 3.

## 4 EXPERIMENTAL SETTING AND RESULTS

**Models.** We evaluate multiple reasoning models from different model families and sizes to test EAGER in comparison to the FULL PARALLEL sampling baseline: SmolLM-3B (HuggingFaceTB, 2025), Qwen3-4B (Team, 2025), DeepSeek-R1-0528-Qwen3-8B (DeepSeek-AI, 2025) and GPT-oss 20B (OpenAI, 2025). Additional generation parameters and EAGER hyper-parameters are available in Appendix D.

**Benchmarks.** We evaluate our approach saved resources (compute metrics) for generation and performance on a set of diverse reasoning benchmarks on various tasks: AIME 2024 and 2025, and the 2025 Harvard MIT Math Tournament (Balunović et al., 2025) for math, GPQA-Diamond (Rein et al., 2023) for scientific domains, and HumanEval Plus (Liu et al., 2023; 2024) for code generation.

**Compute metrics.** We evaluate efficiency improvements using two complementary metrics: The first is the average **sequence Count** (#Seq). FULL PARALLEL sampling uses a fixed budget of $M$ sequences, in contrast, EAGER uses a dynamic #Seq that depends on the branching behavior. The second metric is the average **token Count** (#Token) generated. While #Seq provides a general measure of computational efficiency, #Tokens is a more precise indicator since, branching at step $t$, reuses previously generated tokens $(0, \ldots, t-1)$ as prefix across new branches rather than regenerating them, that can lead to substantial savings even when #Seq is comparable.

**Performance metrics.** We evaluate performance using three complementary metrics. **Pass@k** shows whether the model produces at least one correct final solution, **Cons@k** aggregates responses through majority voting across $k$ generations. Lastly, **Pass Rate** measures the proportion of correct answers over all generated outputs. We report the average metric across each entire benchmark.

### 4.1 RESULTS

**EAGER-init and EAGER yield significant savings in computation.** Figure 3 (top row) illustrates the efficiency advantages of EAGER-init and EAGER across all benchmarks and model scales. Starting with EAGER-init, the total number of generated tokens is typically less than half of that required by FULL PARALLEL sampling. Building on this, EAGER (and EAGER-adapt) leverages a small fraction of the saved budget to further improve accuracy, while still generating substantially fewer sequences than FULL PARALLEL sampling. On the performance side, EAGER consistently achieves higher Pass Rate accuracy than FULL PARALLEL sampling, indicating superior performance per unit of computation. It is worth noting that the performance of SmolLM 3B is 0.0 across all metrics in the parallel-sampling setting. This is caused by the generation of sequences in which the same tokens are repeatedly produced (e.g., "The answer is: The answer is: The ..."). This effect, along with the effect of temperature is discussed in Appendix C.

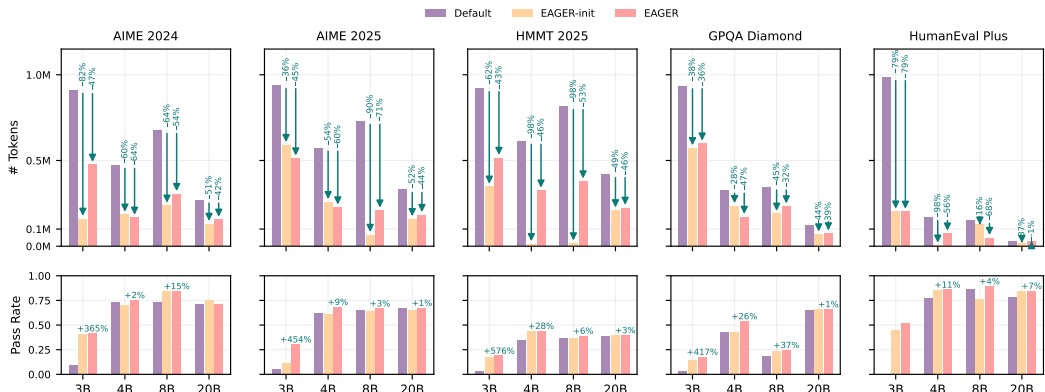

Figure 3: Compute and performance trade-offs of EAGER-init and EAGER. Across all benchmarks and model size, the efficiency of EAGER-init and EAGER consistently outperforms FULL PARALLEL sampling, requiring only half as many tokens in most cases (top). In addition, they achieve higher pass rate accuracy (bottom). For issues specific to the smallest 3B model, see Appendix C.

| Data | | FP | SmolLM 3B EAGER init | adapt | full | FP | Qwen3 4B EAGER init | adapt | full | FP | DeepSeek 8B EAGER init | adapt | full | FP | GPT-Oss 20B EAGER init | adapt | full |
|---|---|---|---|---|---|---|---|---|---|---|---|---|---|---|---|---|---|
| AIME 2024 | ↑p@k | 0.60 | 0.53 | 0.63 | **0.83** | **0.90** | 0.80 | 0.85 | **0.90** | **0.93** | 0.90 | 0.90 | **0.93** | 0.93 | 0.93 | 0.95 | **1.00** |
| | ↓# T | 27 | **18** | 19 | 19 | 14 | **6** | 6 | 6 | 20 | **7** | 9 | 9 | 8 | **4** | 5 | 5 |
| AIME 2025 | ↑p@k | 0.53 | 0.53 | 0.60 | **0.73** | **0.80** | 0.77 | 0.80 | **0.83** | **0.80** | 0.73 | 0.77 | **0.80** | 0.90 | 0.90 | 0.95 | **0.97** |
| | ↓# T | 28 | 19 | **15** | 15 | 17 | **8** | 12 | 12 | 22 | **8** | 13 | 12 | 10 | **5** | 5 | 5 |
| HMMT 2025 | ↑p@k | 0.23 | 0.33 | 0.33 | **0.40** | 0.50 | 0.43 | 0.47 | **0.53** | **0.57** | 0.43 | 0.50 | **0.57** | 0.63 | **0.70** | 0.70 | 0.70 |
| | ↓# T | 28 | 20 | **15** | 15 | 18 | **7** | 15 | 14 | 24 | **8** | 15 | 15 | 13 | **6** | 6 | 6 |
| GPQA-Dia. | ↑p@k | 0.49 | 0.68 | 0.76 | **0.85** | 0.75 | 0.79 | 0.81 | **0.85** | 0.95 | 0.93 | 0.93 | **0.96** | 0.96 | 0.93 | 0.97 | **0.99** |
| | ↓# T | 185 | 137 | 133 | **119** | 65 | **46** | 52 | 59 | 68 | **37** | 42 | 43 | 24 | **13** | 14 | 15 |
| HE-Plus | ↑p@k | 0.00 | 0.04 | 0.68 | **0.75** | 0.91 | 0.86 | 0.92 | **0.94** | 0.95 | 0.96 | 0.96 | **0.98** | 0.95 | 0.96 | **0.97** | 0.97 |
| | ↓# T | 161 | 94 | 94 | **20** | 27 | **10** | 12 | 12 | 25 | 21 | **13** | 13 | 5 | **4** | 5 | 5 |

Table 2: Reallocation of the additional budget (EAGER-adapt) only on Saturating prompts (i.e., prompts that reach $M = 32$ generated sequences). All experiments use a threshold of 2.0, which we found to provide a good balance between number of tokens (# T $\times$ 1e5) used and performance (p@k) across models and benchmarks. **Bold** are best results, underline second best.

**Saturation is a good proxy for budget reallocation.** In the absence of target labels, EAGER-adapt reallocates additional budget to saturating prompts as a proxy for identifying challenging cases. Table 2 reports results across all benchmarks and models[6]. On average, this strategy not only achieves substantial token savings during generation, but also improves exploration, yielding higher Pass@k compared to the FULL PARALLEL sampling baseline. In other words, EAGER-adapt saves a large fraction of compute while at the same time uncovering more successful reasoning paths.

For comparison, Table 2 also reports the full EAGER approach. While it benefits from the unfair advantage of access to target labels, redirecting budget to saturating sequences still achieves the second-best performance in most settings.

**EAGER always achieves better performances than FULL PARALLEL sampling.** As shown in Figure 3, EAGER consistently outperforms FULL PARALLEL sampling in terms of Pass Rate. Table 1 shows a more comprehensive overview using Pass@k, Cons@k, and Pass Rate. While Pass@k is highest under EAGER, Pass Rate is consistently equal or better even for EAGER-init compared to FULL PARALLEL sampling. This suggests that EAGER-init effectively prunes unproductive generations (higher Pass Rate) at the cost of reduced exploration (lower Pass@k). In general, Pass@k is

---

[6]The EAGER-init results in Table 2 differ from those reported elsewhere because we fix the threshold to $\theta = 2.0$ for the EAGER-adapt experiments. In this context, EAGER-init is treated as the first step of the overall process. In other sections, EAGER-init is evaluated as an independent decoding strategy with its optimal threshold selected for both efficiency and performance (see Section 4.1 for details and Appendix D for threshold selection transparency).

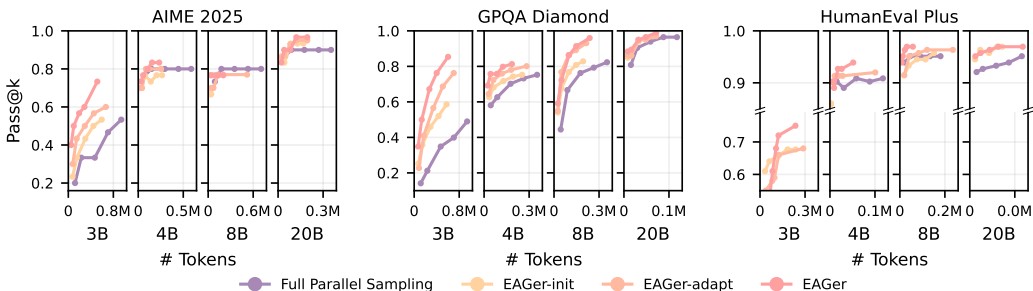

Figure 4: Performance comparison with scaling the total allowed sequences for generating ($M \in \{1, 4, 8, 16, 24, 32\}$). As $M$ increases (line's markers), EAGER consistently improves Pass@k (y-axis) while reducing the number of tokens needed to find the correct solution (x-axis), further shifting the Pareto frontier of the performance–efficiency trade-off.

particularly useful in scenarios where obtaining at least one correct answer is critical, for example, when the user prioritizes correctness and exploration over efficiency as per in Reinforcement Learning applications. In contrast, Pass Rate and Cons@k capture a different dimension of quality: (i) higher values indicate that EAGER focuses computation more effectively on promising generations, and (ii) given the extreme efficiency gains of EAGER-init compared to FULL PARALLEL sampling, the trade-off is often strongly favorable.

**EAGER scales effectively under budget constrains.** We evaluate the effect of scaling the maximum number of allowed generations, $M$, on overall performance. As shown in Figure 4, increasing $M$ improves the probability of obtaining at least one correct solution (Pass@k). This trend is expected, as a larger generation budget naturally enables more extensive exploration. Notably, EAGER-init – and even more so EAGER – achieve superior Pass@k under the same constraints, often with significantly fewer tokens. In other words, EAGER not only benefits from larger $M$ but also allocates its computational budget more efficiently, resulting in a consistent shift of the Pareto frontier, where higher accuracy is achieved at lower token cost.

**Threshold guides the trade-off between performance and compute.** Efficiency metrics (# Tokens, # Seq) are directly shaped by the choice of entropy threshold $\theta$. In our experiments, we explore values in the interval $[1.8, 2.7]$, which captures the majority of observed entropy peaks (see Section 2.2). Across different model families and sizes, we find consistent efficiency improvements relative to the FULL PARALLEL sampling baseline throughout this range. The optimal setting of $\theta$ remains task- and model-dependent. Under EAGER-init, lower thresholds encourage more frequent branching, which increases both the number of generated sequences (#Seq) and total tokens (#Token). Higher thresholds, in contrast, restrict branching, yielding fewer continuations and lower computational cost. The balance between these regimes varies across architectures, scales, and datasets. Full results are available in Appendix A.

## 5 RELATED WORKS

Since the recent introduction of test-time scaling (Snell et al., 2025; Welleck et al., 2024), multiple approaches have been proposed to improve its efficiency and performance. Wu et al. (2025) propose REBASE (REward BAlanced SEarch) a branching method that expands reasoning trajectories that are evaluated as being of high quality by a reward model. While powerful, REBASE is significantly more computationally expensive compared to directly computing token entropy at test-time. Deep-Conf (Deep Think with Confidence, Fu et al., 2025) is a method that also leverages local confidence measures to increase performance and efficiency during generation. DeepConf uses this confidence measure to truncate sequences where it is lower than a pre-defined threshold (determined during a warm-up stage). The combination of this warm-up stage and the truncation of non-promising sequences results in a theoretical token usage that is greater than in our proposed approach, where the certainty measure drives branching instead of truncation.

In Appendix B, we provide a detailed empirical comparison between our approach, DeepConf, and the ESC method (Li et al., 2024). ESC adopts a token-saving strategy based on generating in small

sequential windows and halting early when all trajectories converge to the same final answer, thereby reducing unnecessary continuation.

We note additional works on uncertainty estimation such as Kang et al. (2025) which introduces *self-certainty*, a sequence-level measure closely related to cross-entropy. The authors demonstrate that self-certainty discriminates well between correct and incorrect answers and is robust to reasoning length. The authors additionally illustrate that self-certainty driven answer selection (through a voting mechanism) leads to improvements in reasoning benchmarks. While the the current work is closely related to the work by Kang et al., we demonstrate that token-level certainty (in contrast to sequence-level) can function as a useful tool to modulate performance and efficiency in reasoning LLMs.

## 6 CONCLUSION AND FUTURE DIRECTIONS

By leveraging token-level entropy, EAGER-init proves to be a highly performant training-free generation method with significantly higher efficiency compared to FULL PARALLEL sampling. In applications such as RLVR, where the correct answer is known, EAGER surpasses the Pass@k performance by up to 37% compared to FULL PARALLEL sampling while using up to 65% fewer tokens. Even without a verifier at test time, EAGER-adapt improves exploration, surpassing FULL PARALLEL sampling performance while generating up to 40% fewer tokens. In both cases, the methods demonstrate the ability to *save large amounts of compute and simultaneously enhance exploration*. Finally, we show that the approaches are domain (math, science and coding) and temperature (see Appenidx C) agnostic.

While our current work uses token-level entropy to create branching reasoning streams, future research could explore other methods for quantifying uncertainty. For example, using Kullback-Leibler (KL) Divergence to measure token uncertainty is a promising direction, inspired by the work of Kang et al. (2025). At the same time, a key consideration is that the uncertainty quantification method must be lightweight, as a computationally expensive approach would undermine the goal of improving generation efficiency.

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

# A COMPLETE RESULTS

Table 3 presents a complete overview of the results of our experiments.

| θ | SmolLM3-3B ↑p@1 | ↑c@32 | ↑PR | ↓#T | ↓#S | Qwen3-4B ↑p@1 | ↑c@32 | ↑PR | ↓#T | ↓#S | Deepseek 8B ↑p@1 | ↑c@32 | ↑PR | ↓#T | ↓#S | GPT-oss 20B ↑p@1 | ↑c@32 | ↑PR | ↓#T | ↓#S |
|---|---|---|---|---|---|---|---|---|---|---|---|---|---|---|---|---|---|---|---|---|
| **AIME 2024 (math)** | | | | | | | | | | | | | | | | | | | | |
| — | 0.60 | 0.03 | 0.09 | 27 | 32.0 | 0.90 | 0.80 | 0.74 | 14 | 32.0 | 0.93 | 0.80 | 0.73 | 20 | 32.0 | 0.93 | 0.80 | 0.71 | 8 | 32.0 |
| 2.0 | 0.53 | 0.03 | 0.10 | 18 | 28.8 | 0.80 | 0.73 | 0.70 | 6 | 15.3 | 0.90 | 0.87 | 0.85 | 7 | 15.8 | 0.93 | 0.77 | 0.70 | 4 | 30.7 |
| 2.2 | 0.53 | 0.10 | 0.15 | 18 | 28.4 | 0.70 | 0.67 | 0.68 | 1 | 1.7 | 0.80 | 0.80 | 0.79 | 2 | 4.5 | 0.90 | 0.80 | 0.70 | 4 | 29.7 |
| 2.3 | 0.52 | 0.10 | 0.18 | 17 | 26.8 | 0.67 | 0.67 | 0.65 | 0.3 | 1.0 | 0.77 | 0.77 | 0.76 | 0.4 | 1.3 | 0.90 | 0.80 | 0.68 | 4 | 30.2 |
| 2.4 | 0.67 | 0.20 | 0.25 | 14 | 22.7 | 0.77 | 0.77 | 0.77 | 1 | 1.0 | 0.70 | 0.70 | 0.70 | 0.3 | 1.0 | 0.93 | 0.83 | 0.75 | 4 | 27.6 |
| 2.5 | 0.57 | 0.50 | 0.40 | 5 | 16.5 | 0.73 | 0.73 | 0.73 | 0.2 | 1.0 | 0.73 | 0.73 | 0.73 | 0.3 | 1.0 | 0.90 | 0.83 | 0.69 | 4 | 28.0 |
| 2.0 | 0.73 | 0.10 | 0.13 | 19 | 32.0 | 0.90 | 0.80 | 0.74 | 9 | 23.4 | 0.93 | 0.87 | 0.85 | 9 | 20.0 | 0.93 | 0.77 | 0.70 | 5 | 32.0 |
| 2.2 | 0.73 | 0.10 | 0.16 | 19 | 32 | 0.87 | 0.77 | 0.76 | 5 | 11.3 | 0.90 | 0.80 | 0.80 | 5 | 12.0 | 0.93 | 0.83 | 0.71 | 5 | 32.2 |
| 2.3 | 0.83 | 0.17 | 0.18 | 19 | 33.0 | 0.90 | 0.80 | 0.75 | 5 | 12.4 | 0.87 | 0.83 | 0.81 | 5 | 10.1 | 0.93 | 0.80 | 0.68 | 5 | 32.1 |
| 2.4 | 0.77 | 0.20 | 0.28 | 19 | 32.8 | 0.83 | 0.80 | 0.79 | 5 | 10.5 | 0.90 | 0.80 | 0.78 | 6 | 13.0 | 0.97 | 0.83 | 0.75 | 5 | 32.0 |
| 2.5 | 0.67 | 0.50 | 0.42 | 14 | 31.3 | 0.87 | 0.83 | 0.81 | 5 | 10.7 | 0.90 | 0.77 | 0.77 | 7 | 14.0 | 1.00 | 0.87 | 0.71 | 5 | 32.0 |
| **AIME 2025 (math)** | | | | | | | | | | | | | | | | | | | | |
| — | 0.53 | 0.00 | 0.06 | 28 | 32.0 | 0.80 | 0.70 | 0.62 | 17 | 32.0 | 0.80 | 0.67 | 0.65 | 22 | 32.0 | 0.90 | 0.83 | 0.67 | 10 | 32.0 |
| 1.8 | - | - | - | - | - | 0.77 | 0.70 | 0.60 | 0.90 | 24.5 | - | - | - | - | - | - | - | - | - | - |
| 2.0 | 0.53 | 0.00 | 0.05 | 19 | 28.8 | 0.77 | 0.70 | 0.61 | 8 | 18.4 | 0.73 | 0.60 | 0.59 | 8 | 17.4 | 0.90 | 0.83 | 0.66 | 5 | 31.1 |
| 2.2 | 0.43 | 0.00 | 0.07 | 19 | 29.9 | 0.67 | 0.63 | 0.64 | 1 | 3.1 | 0.70 | 0.63 | 0.64 | 2 | 4.9 | 0.93 | 0.80 | 0.67 | 5 | 30 |
| 2.3 | 0.37 | 0.10 | 0.14 | 17 | 27.8 | 0.60 | 0.60 | 0.60 | 0.3 | 1.1 | 0.70 | 0.67 | 0.66 | 1 | 2.2 | 0.93 | 0.73 | 0.64 | 5 | 31.4 |
| 2.4 | 0.53 | 0.07 | 0.11 | 17 | 27.6 | 0.70 | 0.70 | 0.70 | 0.3 | 1.1 | 0.63 | 0.60 | 0.61 | 0.5 | 1.2 | 0.93 | 0.80 | 0.66 | 5 | 29.8 |
| 2.5 | 0.43 | 0.23 | 0.26 | 10 | 17.5 | 0.60 | 0.60 | 0.60 | 0.3 | 1.0 | 0.63 | 0.60 | 0.61 | 0.4 | 1.2 | 0.90 | 0.83 | 0.69 | 5 | 26.1 |
| 2.0 | - | - | - | - | - | - | - | - | - | - | - | - | - | - | - | - | - | - | - | - |
| 1.8 | - | - | - | - | - | 0.80 | 0.70 | 0.63 | 13 | 32.8 | - | - | - | - | - | - | - | - | - | - |
| 2.0 | 0.63 | 0.00 | 0.06 | 20 | 32.0 | 0.83 | 0.73 | 0.63 | 12 | 30.0 | 0.80 | 0.63 | 0.63 | 12 | 28.0 | 0.90 | 0.83 | 0.66 | 5 | 32.0 |
| 2.2 | 0.57 | 0.00 | 0.08 | 20 | 32.0 | 0.80 | 0.70 | 0.70 | 6 | 14.7 | 0.80 | 0.63 | 0.68 | 7 | 16.1 | 0.97 | 0.80 | 0.68 | 5 | 32.9 |
| 2.3 | 0.57 | 0.10 | 0.15 | 19 | 32.1 | 0.80 | 0.77 | 0.71 | 8 | 17.5 | 0.80 | 0.73 | 0.71 | 6 | 13.1 | 0.93 | 0.73 | 0.64 | 5 | 32.2 |
| 2.4 | 0.67 | 0.07 | 0.13 | 19 | 32.2 | 0.80 | 0.73 | 0.71 | 5 | 10.7 | 0.80 | 0.70 | 0.68 | 7 | 15.3 | 0.93 | 0.80 | 0.66 | 6 | 33.0 |
| 2.5 | 0.73 | 0.33 | 0.31 | 15 | 33.6 | 0.83 | 0.73 | 0.69 | 7 | 15.0 | 0.80 | 0.70 | 0.68 | 6 | 14.0 | 0.93 | 0.83 | 0.69 | 7 | 32.0 |
| **HMMT (math)** | | | | | | | | | | | | | | | | | | | | |
| — | 0.23 | 0.00 | 0.03 | 28 | 32.0 | 0.50 | 0.37 | 0.34 | 18 | 32.0 | 0.57 | 0.43 | 0.37 | 24 | 32.0 | 0.63 | 0.53 | 0.38 | 13 | 32.0 |
| 1.8 | 0.23 | 0.03 | 0.06 | 20 | 31.7 | 0.43 | 0.37 | 0.33 | 10 | 22.7 | - | - | - | - | - | - | - | - | - | - |
| 2.0 | 0.33 | 0.03 | 0.07 | 20 | 30.8 | 0.43 | 0.37 | 0.35 | 7 | 15.5 | 0.43 | 0.37 | 0.33 | 8 | 18.1 | 0.70 | 0.43 | 0.37 | 6 | 31.9 |
| 2.2 | 0.23 | 0.00 | 0.06 | 18 | 28.5 | 0.37 | 0.37 | 0.36 | 1 | 3.3 | 0.40 | 0.37 | 0.38 | 3 | 6.2 | 0.67 | 0.43 | 0.36 | 6 | 32.0 |
| 2.3 | 0.27 | 0.10 | 0.12 | 17 | 27.5 | 0.40 | 0.40 | 0.40 | 1 | 2.3 | 0.40 | 0.40 | 0.35 | 2 | 4.7 | 0.63 | 0.47 | 0.40 | 6 | 30.9 |
| 2.4 | 0.27 | 0.17 | 0.18 | 14 | 23.8 | 0.33 | 0.33 | 0.33 | 0.4 | 1.1 | 0.37 | 0.37 | 0.35 | 1 | 1.5 | 0.63 | 0.43 | 0.38 | 6 | 30.8 |
| 2.5 | 0.23 | 0.17 | 0.17 | 10 | 17.4 | 0.43 | 0.43 | 0.43 | 0.3 | 1.0 | 0.37 | 0.37 | 0.37 | 0.4 | 1.1 | 0.67 | 0.53 | 0.40 | 6 | 30.5 |
| 2.0 | - | - | - | - | - | - | - | - | - | - | - | - | - | - | - | - | - | - | - | - |
| 1.8 | 0.27 | 0.03 | 0.06 | 20 | 32.0 | 0.47 | 0.37 | 0.33 | 15 | 32.9 | - | - | - | - | - | - | - | - | - | - |
| 2.0 | 0.40 | 0.03 | 0.09 | 20 | 32.0 | 0.53 | 0.37 | 0.36 | 14 | 32.0 | 0.57 | 0.40 | 0.35 | 15 | 32.1 | 0.70 | 0.43 | 0.37 | 6 | 32.0 |
| 2.2 | 0.33 | 0.00 | 0.06 | 19 | 32.0 | 0.53 | 0.37 | 0.37 | 11 | 24.5 | 0.57 | 0.43 | 0.43 | 13 | 27.5 | n/a | n/a | n/a | n/a | n/a |
| 2.3 | 0.33 | 0.10 | 0.13 | 19 | 32.0 | 0.57 | 0.40 | 0.41 | 10 | 22.6 | 0.53 | 0.43 | 0.38 | 10 | 24.0 | 0.67 | 0.47 | 0.40 | 7 | 32.0 |
| 2.4 | 0.40 | 0.17 | 0.19 | 17 | 32.2 | 0.50 | 0.40 | 0.37 | 11 | 25.2 | 0.57 | 0.43 | 0.39 | 11 | 24.0 | 0.63 | 0.43 | 0.38 | 7 | 32.0 |
| 2.5 | 0.40 | 0.17 | 0.19 | 15 | 32.7 | 0.60 | 0.43 | 0.44 | 11 | 21.9 | 0.50 | 0.37 | 0.39 | 11 | 23.5 | 0.67 | 0.53 | 0.40 | 7 | 32.0 |
| **GPQA-Diamond (science)** | | | | | | | | | | | | | | | | | | | | |
| — | 0.49 | 0.00 | 0.03 | 185 | 32.0 | 0.75 | 0.51 | 0.43 | 65 | 32.0 | 0.95 | 0.15 | 0.18 | 68 | 32.0 | 0.96 | 0.68 | 0.65 | 24 | 32.0 |
| 2.0 | 0.68 | 0.04 | 0.09 | 137 | 30.8 | 0.79 | 0.51 | 0.43 | 46 | 26.8 | 0.93 | 0.25 | 0.24 | 37 | 28.5 | 0.93 | 0.72 | 0.66 | 13 | 29.7 |
| 2.2 | 0.62 | 0.07 | 0.11 | 130 | 30.0 | 0.71 | 0.47 | 0.43 | 38 | 22.1 | 0.91 | 0.18 | 0.22 | 33 | 25.0 | 0.97 | 0.71 | 0.66 | 14 | 27.6 |
| 2.3 | 0.61 | 0.02 | 0.07 | 133 | 30.7 | 0.64 | 0.49 | 0.43 | 23 | 14.7 | 0.81 | 0.21 | 0.18 | 23 | 17.4 | 0.95 | 0.66 | 0.65 | 14 | 30.9 |
| 2.4 | 0.61 | 0.06 | 0.10 | 126 | 29.8 | 0.56 | 0.45 | 0.44 | 12 | 7.4 | - | - | - | - | - | 0.94 | 0.70 | 0.66 | 14 | 25.8 |
| 2.5 | 0.59 | 0.10 | 0.15 | 113 | 27.6 | 0.49 | 0.46 | 0.45 | 3 | 2.5 | 0.66 | 0.20 | 0.21 | 4 | 3.1 | 0.95 | 0.68 | 0.65 | 13 | 24.0 |
| 2.0 | 0.79 | 0.04 | 0.09 | 137 | 32.0 | 0.85 | 0.51 | 0.43 | 59 | 32.4 | 0.96 | 0.25 | 0.26 | 43 | 32.5 | 0.99 | 0.72 | 0.66 | 15 | 32.3 |
| 2.2 | 0.81 | 0.07 | 0.12 | 132 | 32.0 | 0.81 | 0.50 | 0.45 | 55 | 32.4 | 0.97 | 0.18 | 0.25 | 43 | 33.8 | 0.98 | 0.71 | 0.66 | 15 | 29.9 |
| 2.3 | 0.75 | 0.03 | 0.09 | 135 | 32.0 | 0.81 | 0.53 | 0.47 | 44 | 27.3 | 0.97 | 0.25 | 0.25 | 46 | 35.4 | 0.97 | 0.66 | 0.65 | 14 | 30.9 |
| 2.4 | 0.79 | 0.08 | 0.12 | 128 | 32.0 | 0.81 | 0.51 | 0.50 | 37 | 22.4 | - | - | - | - | - | 0.99 | 0.70 | 0.66 | 15 | 29.9 |
| 2.5 | 0.85 | 0.12 | 0.18 | 119 | 32.1 | 0.81 | 0.59 | 0.54 | 34 | 19.9 | 0.94 | 0.26 | 0.33 | 45 | 35.6 | 0.98 | 0.68 | 0.66 | 14 | 27.3 |
| **HumanEval Plus (code)** | | | | | | | | | | | | | | | | | | | | |
| — | 0.00 | 0.00 | 0.00 | 161 | 32.0 | 0.91 | 0.82 | 0.78 | 27 | 32.0 | 0.95 | 0.90 | 0.86 | 25 | 32.0 | 0.95 | 0.83 | 0.79 | 5 | 32.0 |
| 1.8 | - | - | - | - | - | 0.87 | 0.76 | 0.76 | 16 | 9.9 | 0.95 | 0.82 | 0.79 | 18 | 25.0 | - | - | - | - | - |
| 2.0 | 0.04 | 0.01 | 0.01 | 94 | 30.7 | 0.86 | 0.79 | 0.80 | 10 | 6.20 | 0.96 | 0.85 | 0.77 | 21 | 23.0 | 0.96 | 0.88 | 0.83 | 4 | 24.7 |
| 2.2 | 0.04 | 0.01 | 0.01 | 65 | 26.3 | 0.86 | 0.86 | 0.86 | 1 | 1.1 | 0.94 | 0.86 | 0.83 | 13 | 14.0 | 0.97 | 0.88 | 0.82 | 3 | 23.3 |
| 2.3 | 0.68 | 0.46 | 0.44 | 33 | 17.3 | 0.84 | 0.82 | 0.82 | 0.5 | 1.1 | 0.87 | 0.82 | 0.80 | 6 | 7.4 | 0.93 | 0.81 | 0.74 | 3 | 22.8 |
| 2.4 | 0.52 | 0.37 | 0.38 | 13 | 9.0 | 0.81 | 0.81 | 0.81 | 0.5 | 1.1 | 0.92 | 0.88 | 0.88 | 3 | 3.5 | 0.97 | 0.88 | 0.85 | 3 | 20.3 |
| 2.5 | 0.52 | 0.48 | 0.47 | 3 | 2.6 | 0.82 | 0.82 | 0.82 | 0.4 | 1.0 | 0.88 | 0.87 | 0.86 | 1 | 1.5 | 0.95 | 0.86 | 0.82 | 3 | 18.6 |
| 1.8 | - | - | - | - | - | - | - | - | - | - | - | - | - | - | - | - | - | - | - | - |
| 2.0 | - | - | - | - | - | 0.94 | 0.79 | 0.82 | 17 | 11.2 | 0.97 | 0.77 | 0.74 | 22 | 24.7 | 0.97 | 0.89 | 0.84 | 5 | 29.3 |
| 2.2 | - | - | - | - | - | 0.92 | 0.86 | 0.87 | 9 | 5.7 | 0.96 | 0.87 | 0.83 | 15 | 15.9 | 0.97 | 0.88 | 0.82 | 5 | 29.0 |
| 2.3 | - | - | - | - | - | 0.92 | 0.87 | 0.86 | 11 | 9.9 | 0.98 | 0.88 | 0.86 | 13 | 16.9 | 0.97 | 0.90 | 0.84 | 5 | 28.8 |
| 2.4 | 0.75 | 0.52 | 0.56 | 20 | 19.3 | 0.94 | 0.87 | 0.86 | 12 | 10.9 | 0.97 | 0.90 | 0.89 | 8 | 8.5 | 0.97 | 0.89 | 0.85 | 5 | 26.4 |
| 2.5 | - | - | - | - | - | 0.93 | 0.86 | 0.86 | 12 | 11.8 | 0.96 | 0.91 | 0.90 | 8 | 9.3 | 0.97 | 0.88 | 0.83 | 5 | 27.2 |

Table 3: All models, benchmarks and entropy-thresholds θ configurations. Higher is better for Pass@k (p@k), Cons@k (c@32) and Pass Rate (PR); lower is better for # Token. # Token are in 1e5 unit. Results for FULL PARALLEL sampling generations, EAGER-init generations, and full EAGER. **Bold** is best overall, underline is best within each category always including the FULL PARALLEL sampling one.

## B  RELATED APPROACHES

We include results for the AIME 2025 benchmark comparing several approaches: ESC Li et al. (2024), a sampling method designed to reduce token usage relative to FULL PARALLEL sampling; and DeepConf, the concurrent work of Fu et al. (2025), which estimates the model's confidence in its generated sequence and terminating generation when that confidence falls below a threshold. We report these results in Table 4.

We also include an additional metric, wall-clock time, to highlight the practical impact of these approaches. This metric serves only as a broad overall comparison within the context of these experiments, as wall-clock time is highly sensitive to system configuration and environmental conditions and is not fully isolated from external factors.

| Model | AIME 2025 metric | Baseline FULL PARALLEL | Related Works ESC | Related Works DeepConf | Our EAGER-INIT | Our EAGER-adapt | Our EAGER |
|---|---|---|---|---|---|---|---|
| SmolLM 3B | ↑ Pass@k | 0.53 | 0.53 | **0.63** | 0.53 | **0.63** | 0.73 |
| | ↑ Pass Rate | 0.06 | 0.05 | 0.26 | 0.26 | **0.28** | 0.31 |
| | ↓ # Tokens (1e5) | 28 | 28 | 22 | 19 | **15** | 15 |
| | ↓ Wall-clock time | ~11h | ~60h | ~8h | ~**4.5h** | ~6h | ~8h |
| Qwen3 4B | ↑ Pass@k | **0.80** | **0.80** | **0.80** | 0.77 | 0.80 | 0.83 |
| | ↑ Pass Rate | 0.62 | 0.63 | **0.68** | 0.60 | **0.68** | 0.69 |
| | ↓ # Tokens (1e5) | 17 | 11 | 15 | **8** | 12 | 12 |
| | ↓ Wall-clock time | ~12h | ~25h | ~10h | ~**5h** | ~8h | ~9h |
| DeepSeek 8B | ↑ Pass@k | **0.80** | **0.80** | **0.80** | 0.73 | 0.77 | 0.80 |
| | ↑ Pass Rate | 0.65 | 0.66 | **0.70** | 0.59 | 0.68 | 0.71 |
| | ↓ # Tokens (1e5) | 22 | 13 | 19 | **8** | 13 | 12 |
| | ↓ Wall-clock time | ~27h | ~40h | ~24h | ~**10h** | ~15h | ~16h |
| GPT-Oss 20B | ↑ Pass@k | 0.90 | 0.90 | 0.93 | 0.90 | **0.95** | 0.97 |
| | ↑ Pass Rate | 0.67 | **0.69** | 0.68 | 0.66 | 0.68 | 0.68 |
| | ↓ # Tokens (1e5) | 10 | 7 | 7 | 5 | **5** | 5 |
| | ↓ Wall-clock time | ~29h | ~30h | ~20h | ~**13.5h** | ~**13.5h** | ~17h |

Table 4: Results on the AIME 2025 benchmark comparing the baseline, related approaches (ESC and DeepConf), and our proposed variants. We report Pass@k, Pass Rate, number of generated tokens, and wall-clock time. **Bold** values indicate the best performance among all fair-comparison methods, excluding full EAGER, which has access to ground-truth labels at test time.

Implementation-wise, we adopt ESC using the original authors' default parameters. We set the number of windows per generation to $8$ and the maximum budget to $M = 32$. For DeepConf, we use its online variant, which provides the best performance and is the closest to the intended use cases of both their and our proposed sampling approaches. The online variant includes a per-prompt warm-up stage, which we count toward the reported Pass Rante and the total number of generated tokens. We also set its maximum budget to $M = 32$, matching our setup and the other baselines.

Table 4 clearly shows that our strongest method (full EAGER) outperforms every baseline and related approach. However, we exclude full EAGER from direct comparison and instead highlight the best results among all other methods in **bold**. This exclusion is necessary because full EAGER has an inherent and unfair advantage having access to target labels. Under fair comparison, EAGER-adapt emerges as the strongest overall approach, striking a near-optimal balance between performance (Pass@k and Pass Rate) and efficiency (number of tokens generated, # T) across all models.

We also note that ESC exhibits significantly higher wall-clock times compared not only to both our approach and DeepConf but also to FULL PARALLEL sampling. This is a direct consequence of its design: ESC relies on small sequential windows and repeatedly evaluates intermediate results to decide whether to continue to the next window, up to the full budget $M$. While this approach does reduce token usage relative to FULL PARALLEL sampling, it severely restricts parallelization, resulting in considerably slower overall execution.

More broadly, our *branch-and-reuse* strategy provides consistent advantages for several reasons: (i) it reuses previously generated tokens, (ii) avoids producing multiple sequences when the model is already *confident*, (iii) requires no warm-up stage (unlike DeepConf-online) and (iv) KV-cache reloading has negligible overhead at the scale of our long generations, as evidenced by the wall-clock results. Empirically, we achieve comparable or superior performance across similar settings. We also observe that for easier prompts (where Pass Rate $\approx 1$, i.e., all generated answers are identical and correct), our method saves substantial tokens by generating only one or two branches. In

contrast, DeepConf – relying solely on confidence estimates – often judges the model confident enough to generate all 32 sequences, leading to fully redundant token generation.

## C  EFFECT OF TEMPERATURE

The temperature hyperparameter, $\tau$, plays a critical role during autoregressive decoding by scaling the logits used by the sampling method (decoding becomes more greedy as $\tau \to 0$). In this section, we conduct a short exploration on the effect of temperature on EAGER. This is especially important in the current context, where a higher diversity among the generated sequences can intuitively have an effect on the performance metrics. For this exploration, we focus on two LLMs, SmolLM 3B & DeepSeek 8B, two temperature settings, $\tau \in \{0.6, 0.9\}$ and AIME 2025 as the evaluation dataset. Furthermore, we conduct the analysis for varying entropy threshold levels $\theta \in \{2.0, 2.2, 2.3, 2.4, 2.4, 2.5\}$.

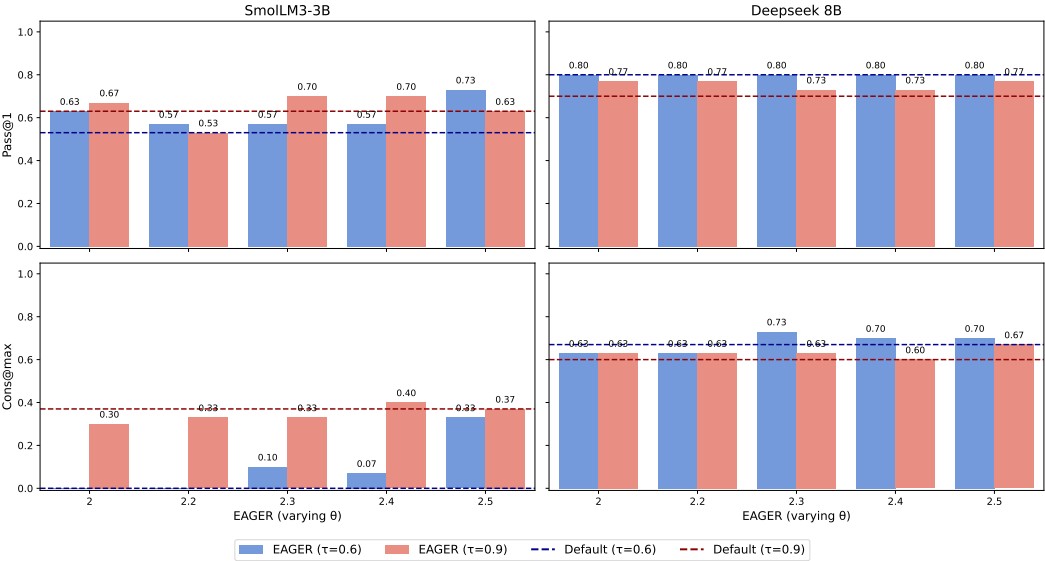

Figure 5: Pass@k and Cons@k at low ($\tau = 0.6$) and high($\tau = 0.6$) temperature settings. Horizontal lines show the performance for the default sampling method, while the bars show EAGER's performance for varying entropy threshold levels $\theta$.

As shown in Figure 5, SmolLM 3B generally performs best at the high temperature setting while the opposite is true for DeepSeek 8B. Importantly, at both temperature levels, EAGER is competitive with the corresponding default baselines, often surpassing them. A direct comparison between the low and high temperature setting including all metrics for default, EAGER and EAGER-init generations is presented in Table 5.

Notably, the performance of SmolLM 3B is particularly higher in the high temperature setting when measured by the Cons@max rate. We find that this is a result of the reduction of generations in which the same tokens are repeatedly produced (e.g., "The answer is: The answer is: The ..."). Specifically, in the high temperature setting, this phenomenon occurs, on average, 59.1% less compared to the low temperature setting. This behaviour was only observed with SmolLM 3B, suggesting it results from the smaller model size. An exception arises with the HumanEval Plus benchmark, where SmolLM 3B failed to solve any tasks, resulting in all metrics being zero under the FULL PARALLEL sampling setting. In contrast, EAGER-init and EAGER appeared to partially mitigate this issue.

Lastly, we also find that the temperature has an effect on the number of tokens generated which, by extension, impact performance. For example, when EAGER is used at the high temperature setting, Deepseek 8B generates, on average, less than half the number of tokens compared to the low temperature setting. In contrast, SmolLM3-3B generates more tokens at the high-temperature setting.

| θ | SmolLM3-3B Low Temperature ($\tau = 0.6$) | | | | | High Temperature ($\tau = 0.9$) | | | | | Deepseek 8B Low Temperature ($\tau = 0.6$) | | | | | High Temperature ($\tau = 0.9$) | | | | |
|---|---|---|---|---|---|---|---|---|---|---|---|---|---|---|---|---|---|---|---|---|
| | ↑p@1 | ↑c@32 | ↑PR | ↓#T | ↓#S | ↑p@1 | ↑c@32 | ↑PR | ↓#T | ↓#S | ↑p@1 | ↑c@32 | ↑PR | ↓#T | ↓#S | ↑p@1 | ↑c@32 | ↑PR | ↓#T | ↓#S |
| - | 0.53 | 0.00 | 0.06 | 28 | 32.0 | 0.63 | 0.37 | 0.25 | 44 | 32.0 | 0.80 | 0.67 | 0.65 | 22.0 | 32.0 | 0.70 | 0.60 | 0.57 | 7.8 | 32.0 |
| 2.0 | 0.53 | 0.00 | 0.05 | 19 | 28.8 | 0.67 | 0.30 | 0.25 | 26 | 31.0 | 0.73 | 0.60 | 0.59 | 8.0 | 17.4 | 0.77 | 0.63 | 0.58 | 3.2 | 22.7 |
| 2.2 | 0.43 | 0.00 | 0.07 | 19 | 29.9 | 0.53 | 0.37 | 0.25 | 27 | 30.9 | 0.70 | 0.63 | 0.64 | 2.0 | 4.9 | 0.70 | 0.60 | 0.58 | 1.9 | 11.3 |
| 2.3 | 0.37 | 0.10 | 0.14 | 17 | 27.8 | 0.57 | 0.33 | 0.26 | 26 | 29.9 | 0.70 | 0.67 | 0.66 | 1.0 | 2.2 | 0.57 | 0.53 | 0.53 | 0.7 | 4.4 |
| 2.4 | 0.53 | 0.07 | 0.11 | 17 | 27.6 | 0.60 | 0.40 | 0.32 | 22 | 25.8 | 0.63 | 0.60 | 0.61 | 0.5 | 1.2 | 0.57 | 0.53 | 0.53 | 0.7 | 4.0 |
| 2.5 | 0.43 | 0.23 | 0.26 | 10 | 17.5 | 0.50 | 0.37 | 0.26 | 21 | 24.7 | 0.63 | 0.60 | 0.61 | 0.4 | 1.2 | 0.63 | 0.60 | 0.61 | 0.3 | 2.0 |
| 2.0 | 0.63 | 0.00 | 0.06 | 20 | 32.0 | 0.67 | 0.30 | 0.25 | 27 | 32.0 | 0.80 | 0.63 | 0.63 | 12.0 | 28.0 | 0.77 | 0.63 | 0.58 | 4.4 | 30.2 |
| 2.2 | 0.57 | 0.00 | 0.08 | 20 | 32.0 | 0.53 | 0.33 | 0.25 | 28 | 32.0 | 0.80 | 0.63 | 0.68 | 7.0 | 16.1 | 0.77 | 0.63 | 0.62 | 3.5 | 21.0 |
| 2.3 | 0.57 | 0.10 | 0.15 | 19 | 32.1 | 0.70 | 0.33 | 0.27 | 28 | 32.1 | 0.80 | 0.73 | 0.71 | 6.0 | 13.1 | 0.73 | 0.63 | 0.63 | 3.2 | 21.1 |
| 2.4 | 0.67 | 0.07 | 0.13 | 19 | 32.2 | 0.70 | 0.40 | 0.33 | 29 | 32.0 | 0.80 | 0.70 | 0.68 | 7.0 | 15.3 | 0.73 | 0.60 | 0.59 | 3.1 | 18.9 |
| 2.5 | 0.73 | 0.33 | 0.31 | 15 | 33.6 | 0.63 | 0.37 | 0.29 | 29 | 32.4 | 0.80 | 0.70 | 0.68 | 6.0 | 14.0 | 0.77 | 0.67 | 0.66 | 2.4 | 14.9 |

Table 5: AIME 2025 results for default, EAGER-init, and EAGER generations for low and high temperature $\tau$ and varying entropy threshold $\theta$. Best results per temperature and threshold setting are marked in **boldface**.

In both cases, and in line with the test-time scaling paradigm, we find that higher performance is achieved in whichever temperature setting more tokens are generated.

# D GENERATION PARAMS

All models are used with their longest thinking configuration to get their best performances. Furthermore we limit their context window to 32k tokens. All sequences are generated with a temperature of $\tau = 0.60$ and a top-p of $95\%$. The effect of temperature is discussed in Appendix C. Table 6 reports the thresholds used for each benchmark and model. Following the discussion in Section 4.1, we select thresholds independently based on their intended use. The EAGER-init sampling method is designed to save budget without significantly compromising performance (lower threshold), whereas EAGER aims to preserve as much performance as possible for later reuse, higher threshold are preferred in such scenario.

| | SmoLM 3B | | Qwen3 4B | | DeepSeek 8B | | GPT-oss 20B | |
|---|---|---|---|---|---|---|---|---|
| | EAGER-init | EAGER | EAGER-init | EAGER | EAGER-init | EAGER | EAGER-init | EAGER |
| AIME 2024 | 2.5 | 2.5 | 2.0 | 2.3 | 2.0 | 2.0 | 2.4 | 2.5 |
| AIME 2025 | 2.4 | 2.5 | 2.0 | 2.5 | 2.2 | 2.5 | 2.4 | 2.5 |
| HMMT 2025 | 2.5 | 2.5 | 2.5 | 2.5 | 2.4 | 2.5 | 2.5 | 2.5 |
| GPQA-Diamond | 2.5 | 2.5 | 2.0 | 2.5 | 2.0 | 2.3 | 2.2 | 2.0 |
| HumanEval Plus | 2.3 | - | 2.2 | 2.4 | 2.0 | 2.4 | 2.4 | 2.4 |

Table 6: Best thresholds for every benchmark and model.

