# OpenReview forum: "EAGER: Entropy-Aware GEneRation for Adaptive Inference-Time Scaling"
_ICLR.cc/2026/Conference — Submitted to ICLR 2026_

### Official Review · Reviewer_XMDw · 2025-10-30

**Soundness:** 2
**Presentation:** 1
**Contribution:** 3
**Rating:** 4
**Confidence:** 3

**Summary:**

An entropy-based branching technique in generation is proposed and evaluated over various datasets. It can achieve better trade-offs by improving the efficacy of test-time scaling. The sampling is also adaptive, adding extra effort to more challenging questions.

**Strengths:**

The proposed entropy-aware generation, where branches depend on entropy, is intuitive and simple.

Experiments cover different datasets in different areas and show the improvement for better trade-offs for computation-accuracy. Also, experiments show that budgets are used adaptively, which is good in practice.

**Weaknesses:**

Figure 1 is not very intuitive and not very helpful to understand the method, despite the method itself is not complicated.

EAGER results need ground truth, so it is like a theoretical upper bound. So, for clarity, the tables should make this clearer, like adding footnotes to EAGER. EAGER-init and +budget should be the practical ones, useful for practice instead.

Focusing on two mathematics datasets in Table 2 rather than the same datasets as in Table 1 is not clearly motivated. Why are these two separate experiments instead of a comprehensive experiment comparing baseline, EAGER-init, +budget, and EAGER? Also, why is Figure 3 the union of the datasets of Tables 1 and 2 plus AIME 2024? It would be clearer if different tables/figures were on consistent datasets and represented different perspectives for holistic comparison; in such a case, the conveyed information would overlap.

Minor: More precision in Table 2, especially for the number of tokens, is recommended.

**Questions:**

Is it possible that the branching points concentrate on the early part of the generation in the case of saturation, so that the branches happen too early? In this case, the latter places where branches are also useful may be ignored. In the case where the challenging question has multiple parts and your method saturates, will diversity/correctness be biased to the earlier parts?

Do people need to tune M for each dataset / how generalizable is it? Is it just a trade-off, or are there still optimal ranges?

Can you explain the meaning of the line in Figure 2? It does not look like the trend line.

Is it possible to further improve branching by distinguishing useful entropy (where randomness is more essential/relevant to the problem) vs empty entropy?

Is it possible to make the entropy threshold soft? Is there an optimal strategy?

How is “saves up to 80%” estimated?

---

> ### Author Response · Authors · 2025-11-20
>
> Dear Reviewer XMDw,
>
> Thank you for your constructive and detailed review! Below, we address each of your points in detail:
>
> **On figure 1 being not intuitive:**
>
> We thought a lot both before and after submission on Figure 1, and we are continuing to refine it. If there is a specific part that you found particularly unclear, we would be grateful for further guidance. In the meantime, we have made several targeted improvements aimed at enhancing readability and conveying the sequence of events more clearly:
> - Added clarification boxes to highlight high-entropy tokens that trigger branch creation.
> - Improved the distinction between italicized text (representing actual model generations) and the surrounding explanatory text.
> - Added step-based annotations to more clearly illustrate the workflow (EAGer-init on all prompts, followed by reallocation of leftover budget).
>
> **On EAGER results needing ground truth and thus considered as an upper bound:**
>
> We agree that the full EAGer variant assumes access to correctness signals, which makes it effectively an upper-bound configuration. Following similar feedback from other reviewers, we have clarified this distinction throughout the paper and in the updated tables. In particular, EAGer-adapt (formerly EAGer +budget) is now presented as the practical, fully test-time-deployable version of our method, and its results, expanded across all benchmarks, show that it consistently matches the efficiency of EAGer-init while outperforming standard Full Parallel Sampling in accuracy (up to +43% in pass@k for HMMT25 with the smallest model). We explicitly mark EAGer as an oracle setup in tables and captions to avoid confusion.
>
> **On limited set of benchmark of Table 2**
>
> Thank you for pointing out improvements about the presentation. At the time of writing we decided to explicitly focus on mathematical reasoning, thus presenting the two most used mathematical dataset we had in our evaluation set. However, as previously mentioned to other reviewers, we have expanded Table 2 to include all model families and task domains (mathematics, science, and code) under the same unified setup. The updated tables and figures now align consistently across datasets and variants in the updated paper uploaded as a new rebuttal submission.
>
> **On precision of Table 2**
>
> As previously mentioned, we have revised Table 2 to include higher numerical precision and harmonized formatting with other figures and tables.
>
> **Questions:**
>
> - **Does branching happen to early, saturating M?** Indeed, when using very low thresholds (see Appendix Table 3, yellow rows for EAGER-init), the number of generated sequences (#S) approaches the maximum budget M=32, probably indicating early saturation. However, the empirically optimal threshold we select (~2.4, Table 4), leads to a much smaller average #S, ensuring that our budget $M$ doesn’t pose any limit at when branching occurs. Also, empirical inspection shows that branching points are roughly uniformly distributed across generations. We will clarify this behavior in Appendix C and note that analyzing the positional role of branching (early vs. late reasoning) is an interesting direction for future work.
> - **Do I need to tune M?** Yes, though EAGer is far less sensitive to M than standard approaches. Because EAGer adjusts its behavior to the prompt’s perceived complexity, M mainly reflects compute and latency constraints. Our experiments show diminishing returns beyond M=30 and suggest a practical range of 10–30.
> - **What’s the meaning of the line in Figure 2?** The red line in Figure 2 is the fitted linear regression showing the correlation between peak entropy and Pass@k success. We use it to extract the slope and intercept, and will clarify this explicitly in the caption.
> - **Can we use useful/empty entropy to further improve our method?** We appreciate this insightful suggestion. Conceptually, one could distinguish between aleatoric uncertainty (arising from incomplete reasoning context) and epistemic uncertainty (reflecting model indecision) [e.g. 1]. While our current framework does not explicitly separate these types, we agree that doing so could provide valuable interpretability and efficiency gains. We will note this as an important avenue for future research.
> - **Is it possible to make the entropy threshold soft?** We tested adaptive thresholds but found fixed thresholds work robustly across settings. Designing a learned, confidence-conditioned threshold remains an interesting direction for future work.
> - **How is “saves up to 80%” estimated?** Thanks for pointing this out. The 80% figure was a legacy metric based on the number of saved sequences rather than generated tokens. To ensure consistency and clarity, we have removed the 80% claim and now report only the empirically verified 65% token reduction (line 095).
>
> [1] Measuring Aleatoric and Epistemic Uncertainty in LLMs: Empirical Evaluation on ID and OOD QA Task
>
> (edit: formatting)

---

### Official Review · Reviewer_F2Hp · 2025-10-31

**Soundness:** 2
**Presentation:** 3
**Contribution:** 3
**Rating:** 4
**Confidence:** 3

**Summary:**

The paper presents EAGER, a training-free decoding method that adapts compute usage to prompt difficulty using token-wise entropy. It branches reasoning only when uncertainty is high, cutting redundant computation. On reasoning benchmarks like AIME 2025, EAGER reduces token generation by up to 65% while improving accuracy by 27%, achieving a better efficiency–performance trade-off.

**Strengths:**

1. The paper is well-written and easy to follow.
2. The proposed approach is simple and effective. Especially, it always outperforms the full parallel baseline, showcasing that it improves the overall reasoning performance as well as improving the efficiency.
3. The authors conducted extensive experiments using models from diverse families, across diverse scales.

**Weaknesses:**

1. Limited baselines. While comparison against full parallel baseline is valuable, there is a large body of existing works on training-free approaches for budget control [1]. Comparing against at least some of such existing work would be helpful to understand the relative position of the proposed approach in the big picture.

2. Limited efficiency analysis. While the main efficiency metric is the total number of decoded tokens, this may not directly translate to peak memory usage or inference latency, especially in batch generation scenarios where the sequences are generated in parallel. Further analysis regarding these two aspects would strengthen the claims regarding efficiency.

3. Minor formatting issues. Bold is missing for some entries in Table 1. Also, the method name is written as EAGER in the title, but EAGer in the rest of the text.

[1] Sui et. al., Stop Overthinking: A Survey on Efficient Reasoning for Large Language Models, TMLR 2025

**Questions:**

Could you provide some additional analysis regarding additional baselines and efficiency measurements?

---

> ### Author Response · Authors · 2025-11-20
>
> Dear Reviewer F2Hp,
>
> Thank you for your review. We appreciate your recognition of our writing clarity, methodological simplicity, and comprehensive experimental evaluation. We address your first two main points below.
>
> **On a potential baseline expansion:**
>
> We agree that including additional training-free baselines for adaptive budget control would further contextualize EAGer within the broader literature. At submission time, our focus was primarily on the Full-Parallel baseline to isolate the contribution of entropy-guided branching. Since then, we have integrated a few other baselines such as [1, DeepConf] (a concurrent yet different approach to our entropy-branching) and [2, ESC] an already published early-stopping criteria to save budget under test-time scaling setting.
> We present this table in the general comment above and in the updated version of the paper (Appendix B) alongside some consideration based on the added results.
>
> We also note that, generally, approaches mentioned in the cited review and even more recent works [e.g. 3, 4, 5] optimize somewhat different objectives. While the others focus on maximizing probability of correct solution under fixed rollout budget given multiple candidates, our approach (and [1, 2]) focuses on reducing redundant computation across prompts by adaptly allocating resources while generating. Nonetheless, we will explicitly discuss these differences in the revised Related Work section.
>
> [1] Deep Think with Confidence
>
> [2] Escape Sky-high Cost: Early-stopping Self-Consistency for Multi-step Reasoning, ICLR 2024
>
> [3] Every Rollout Counts: Optimal Resource Allocation for Efficient Test-Time Scaling NEURIPS 2025
>
> [4] Make Every Penny Count: Difficulty-Adaptive Self-Consistency for Cost-Efficient Reasoning NAACL 2025
>
> [5] Leveraging Reasoning Paths for Efficient LLM Sampling NAACL 2025
>
>
> **On the limited efficiency analysis:**
>
> We appreciate your suggestion to broaden the efficiency discussion beyond token count. In particular we consider two scenarios:
> 1. the easier one where branching is performed sequentially as explained by Figure 1 (wait for an uncertain token, branch from there and repeat for every active sequence): here, efficiency gains are most evident in total compute and GPU power consumption, as reduced token generation directly lowers active FLOPs and memory pressure.
>
> 2. when branching is performed in parallel (*), efficiency gains are also evident in e.g. wall-clock time (see table above). In practice, this achieves near-constant ~100% GPU occupancy while reducing total wall-clock time proportionally to token savings.
>
> (*) More technically, our prototype implementation of parallel generation is performed as follows: we start by batching $b$ out of the e.g. 30 prompts from AIME and generate the first sequence for all of these, storing indexes of high entropy tokens on the side. Then, we do a second pass filling the current batch with leftover branches from the previous batch(es), plus, new sequences. We repeat this process until no more sequences and no more branches need to be completed. This ensures that the GPU is always at 100% utilization both for memory and occupancy.
> Since the implementation leveraging advanced optimizations techniques such as paged attention is out-of-scope for this paper, we only tested this approach using the default HuggingFace implementation.
>
> **Minor formatting issues:**
> Thank you for catching these. We have corrected the missing boldface entries in Table 1 and standardized the method name to EAGer throughout the paper.
>
> **Question:** addressed previously
>
> Thank you again for your feedback!

---

### Official Review · Reviewer_veE8 · 2025-11-01

**Soundness:** 3
**Presentation:** 3
**Contribution:** 3
**Rating:** 4
**Confidence:** 3

**Summary:**

EAGER monitors token-level entropy during decoding and branches only at high-entropy steps (EAGER-init), reusing the prefix so low-entropy spans are not re-generated. The saved budget is then reallocated to saturating/hard prompts up to a global cap M. The paper reports a negative correlation between peak entropy and pass rate and shows up to 65% fewer tokens with up to +27% Pass@1 vs FULL-PARALLEL across math/science/code tasks and 3B-20B models. Key knobs include the entropy threshold, top-2 temporal branching, a no-branch window, and simple reallocation rules.

**Strengths:**

- Simple, general heuristic: Thresholding token-level entropy to trigger branching is conceptually clean and model-agnostic.
- Empirical signal: The correlation between peak entropy and failure provides a concrete control signal.
- Practicality / deployability: Prefix reuse (KV-cache effectiveness) makes the method easy to integrate into existing parallel sampling paths and likely efficient in real serving stacks.
- Quality–efficiency gains: Reports simultaneous token savings and accuracy lifts across models and budgets.

**Weaknesses:**

- While superiority over FULL-PARALLEL is clear, a head-to-head comparison under equal budget with other adaptive policies—especially DeepConf—would be highly informative. Both methods aim to allocate compute dynamically, yet take opposite strategies: DeepConf terminates low-confidence continuations, whereas EAGER branches on high-entropy points. It would be interesting to see which yields better token reduction, accuracy, and wall-clock efficiency under the same budget. I expect EAGER to have an advantage in runtime due to KV-cache reuse, but empirical evidence would clarify whether “cut-and-restart” or “branch-and-reuse” is ultimately more effective.

- Sensitivity/assumptions: Outcomes may hinge on θ, temperature M, and the top-2 branching choice; difficulty-binned analyses (e.g., overthinking/U-shape on easy items) are missing.

**Questions:**

- Can you report same-budget curves vs DeepConf under (a) equal tokens, (b) equal wall-clock (incl. TTFT), (c) FLOP proxies?
- What is the per-token overhead of entropy checks and branch management in a standard serving stack, and how does KV-cache sharing affect memory/throughput?
- How sensitive are results to θ, M, temperature/top-p? Any auto-tuning or per-prompt adaptation?
- Does top-2 branching trade off diversity vs sampling two tokens independently? Any study with top-k or stochastic branching?
- Can entropy be approximated (e.g., with a tiny probe or confidence surrogate) when logits are unavailable, and how does that impact gains?

---

> ### Author Response · Authors · 2025-11-20
>
> Dear Reviewer veE8,
>
> Thank you for your detailed feedback. We appreciate your recognition of EAGER’s simplicity, empirical grounding, and practical deployability. We address your main concerns and questions below.
>
> **On the inclusion of additional baselines, such as DeepConf:**
>
> We included DeepConf [1] in the related works but not as a baseline only because of time constraints. DeepConf was released only shortly before the ICLR deadline, and it was not possible to complete a robust integration and evaluation within our benchmarking suite. However, we have since updated our experiment suite, which now also includes DeepConf. Specifically, we selected the best and similar DeepConf approach from their paper (named “online”) with an initial warmup stage to get the correct confidence thresholds and set the same maximum number of sequences as in our paper (M=32). We compare their decoding strategy to our EAGer-adapt (newly renamed, formerly known as EAGer + budget). As shown in the table included in the general comment above, we observe a clear advantage of our approach on AIME 2025.
>
> As anticipated in your review, **the “branch-and-reuse” technique is better in most cases for different reasons**: (i) reuses tokens already generated, (ii) avoids the generation of multiple sequences when the model is already “confident” (iii) does not need any warmup stage (applies only to DeepConf online) (iv) KV-cache reloading does not meaningfully impact over such long generations (see wall-clock time). Empirically, we also show that performance is almost always better with our EAGer-adapt method compared to DeepConf across all four models, and the wall-clock time again demonstrates a clear advantage.
>
> We also note that, especially for easier prompts (where Pass Rate ~1, i.e., all the generated answers are the same and all correct) we save significantly more tokens by only generating one or two branches. DeepConf, basing its decision on confidence and being confident at generating all 32 sequences, redundantly generates them all.
>
> We also include ESC [2], as baseline; we were not aware of it at submission time.
>
> We already included these results in the update version of the paper (Appendix B).
>
> [1] Deep Think with Confidence, 2025
>
> [2] Escape Sky-high Cost: Early-stopping Self-Consistency for Multi-step Reasoning, ICLR 2024
>
> **Sensitivity and assumption being threshold- and design choices- dependent:**
>
> We acknowledge that several hyperparameters influence EAGer’s behavior and we provide further clarification: (i) As discussed with Reviewer Rbnr, the optimal threshold is remarkably stable across models and benchmarks (~2.5). This consistency suggests that tuning efforts remain minimal in practice. (ii) We explored $M$ values in [1 to 32], reflecting realistic inference budgets. Figure 4 illustrates the general scaling trend with respect to M. (iii) Top-2 branching as design choice provides a balanced trade-off between exploration and computational control, avoiding exponential branching early in decoding. We plan to explore higher-arity or stochastic branching schemes in future work.
>
>
> Regarding the analysis of response difficulty levels, our understanding is that such semantic evaluations are (typically) performed by external judge models on the reasoning traces. While this type of analysis could provide complementary insights, it is somewhat beyond the scope of this paper primarily focused on efficiency and performance gains in reasoning models. We appreciate the suggestion and see it as a promising direction for future exploration.

---

> > ### Author Response · Authors · 2025-11-20
> >
> > **Questions:**
> > - **More comparison with DeepConf:** We reported wall-clock time in the table above. KV cache reloading incurs negligible time cost compared to a standard generation: even in the worst case (reloading $M-1=31$ times), the cost is dominated by the significantly larger number of generated tokens, and thus has no meaningful impact on end-to-end time.
> > Regarding TTFT, it is virtually 0 for EAGer (if you exclude the model’s loading time from vLLM) since the first generated sequence is immediately usable. By contrast, DeepConf’s “online” variant has a much higher TTFT due to its warmup stage required to collect confidence intervals for each prompt.
> > As for FLOPs, they are practically the same for both approaches. Both methods rely on the same generation APIs and gather per-token statistics during decoding. The underlying compute workload is therefore practically identical.
> > - **Per-token overhead:** Entropy computation is negligible, performed in linear time and implemented within the same step as token sampling, thus adding no measurable latency to generation. KV-cache considerations were addressed above; empirically, cache sharing and reloading do not materially affect throughput or memory use in our setting.
> > - **Sensitivity to $\theta$, M, temperature/top-p?** We show ablations for $M$ (Fig. 4), generation parameters like temperature (Fig. 5) and finally present a comprehensive table with all tested $\theta$ in Table 3. Across these, performance remains robust within reasonable parameter ranges.
> > We explored ideas such as auto-tuning however, empirically the best thresholds obtained via our bin search remain highly stable across models and benchmarks. In practice, we recommend using a default $\theta = 2.5$ and adjusting slightly based on the guidelines in Section 4..
> > - **Diversity under top-2 branching:** We internally tested stochastic vs. top-k branching and observed minimal variation in outcomes. Empirically, branching itself, not the exact token choice, drives diversity and reasoning gains, as it allows exploration of distinct reasoning paths rather than token-level variation. We also note that, most of the high-entropy tokens do not meaningfully indicate a specific reasoning direction, they only offer the model the opportunity to think along a different path (as also noted by related works [3]). We will make this clearer and add notes on possible design extensions in the final version.
> > - **Entropy approximation when logits are unavailable:** In Section 2 we showed the correlation between perceived difficulty and token entropy peaks. We speculate that this correlation can be approximated using lightweight confidence surrogates or probe models (though with higher overhead). Given the possible span of potential information we can get from a model, we leave this investigation to future research.
> >
> > [3] Beyond the 80/20 Rule: High-Entropy Minority Tokens Drive Effective Reinforcement Learning for LLM Reasoning
> >
> > We thank you again for the insightful feedback. We really appreciated a detailed, well- and deep- thought review!

---

### Official Review · Reviewer_Rbnr · 2025-11-01

**Soundness:** 3
**Presentation:** 3
**Contribution:** 3
**Rating:** 6
**Confidence:** 4

**Summary:**

The paper introduces a training-free decoding scheme for test-time scaling called EAGER. So instead of generating M parallel CoT traces per prompt no matter what, it watches token-level entropy and branches only at high-entropy steps, reusing the shared prefix; easy prompts consume 1–2 traces, hard ones branch more. The unused “sequence budget” from easy prompts is then reallocated to prompts that failed Pass@1, so total compute stays within the original M×|D| but gets spent where it matters. Across multiple benchmarks the method showed to be using up to 65% fewer tokens and still beat or match full-parallel sampling, sometimes by as much as +27% Pass@1 on small models.

**Strengths:**

- Clear motivation and definition of the problem
- Training free method which makes it scalable and works across a good range of model sizes.
- Good performance across benchmarks

**Weaknesses:**

- Needs per-token entropy + tuned threshold θ; θ is model/task-dependent, so not totally plug-and-play.
- Reallocation assumes you can tell which prompts failed ( access to answers/verifier); without that, they fall back to a heuristic.
- Entropy is shown to correlate with difficulty on these reasoning tasks, but generality beyond that is argued more than proved in the paper.

**Questions:**

Please refer to weaknesses

---

> ### Author Response · Authors · 2025-11-20
>
> Dear Reviewer Rbnr,
>
> Thank you for your thoughtful feedback! We address your three main concerns below:
>
> **On the threshold $\theta$ being model- or task-dependent:**
>
> While it’s true that the threshold requires tuning for each model and task, empirically we show and discuss (end of section 4) that the optimal threshold is remarkably consistent across our evaluation suite (4 different models, with different sizes, and 5 different benchmarks). While more exhaustive testing could yield marginal improvements, our empirical results indicate that a threshold close to 2.5 consistently performs well for all settings considered (see Appendix C, Table 5). This finding indicates that the method remains largely robust and practical despite this tuning step.
>
> **On the reallocation requiring the target label to allocate additional budget:**
>
> We agree that the full EAGer method assumes access to ground-truth correctness signals to optimally reallocate unused sequence budgets. This assumption holds naturally in certain settings, such as RLVR or test-time verification pipelines, where a verifiable reward signal is available.
>
> Nonetheless, to eliminate the need for test-time labels, we also introduce **EAGer-adapt** (new name introduced for clarity, formerly named EAGer + budget in the paper), a variant designed for cases where target labels are unavailable.
>
> To this end, we extended our experiments to include EAGer -adapt across all models and benchmarks, and the new results, visible in the new update Table 2 in the submitted paper, confirm that it maintains the same efficiency as EAGer-init while consistently surpassing the performance of the standard Full Parallel Sampling approach (e.g. up to +43% in pass@k for HMMT with the smallest model), even without oracle feedback as hinted by the original experiments.
>
> **On the generality of entropy as a proxy for reasoning difficulty:**
>
> We agree that while we empirically demonstrate a strong correlation between token-level entropy and reasoning difficulty, a theoretical justification for this relationship remains an open question. Our primary goal in this work was to show that this empirical correlation can be effectively leveraged for test-time efficiency and performance gains. We have clarified this point in the revised manuscript and explicitly note that a deeper theoretical analysis of the entropy-difficulty relationship would nicely complement the current work.

---

### Author Response · Authors · 2025-11-20

We provide this **general comment to share additional results obtained using further baselines that we believe are broadly relevant to all reviewers and area chairs**.

These results include comparisons with a concurrent work (DeepConf, [1]), released around the ICLR deadline, which is highly pertinent to our approach and was requested by some reviewers to strengthen the overall presentation. In our individual responses to each reviewer, we reference these results and address their specific questions in detail.

The same table is also available in Appendix B of the updated paper.


| Model          | Metric          | FullParallel | ESC [2] | DeepConf [1] | EAGer-init | EAGer-adapt | EAGer(*) |
|----------------|-----------------|--------------:|--------:|-------------:|------------:|-------------:|----------:|
| **SmolLM 3B**  | ↑ pass@k        |   0.53 | 0.53   | **0.63**    | 0.53       | **0.63**        | 0.73 |
|                | ↑ Pass Rate(**)| 0.06         | 0.05   | 0.26        | 0.26       | **0.28**        | 0.31 |
|                | ↓ # Tokens (1e5)  | 28           | 28     | 22          | 19         | **15**      | 15   |
|                | ↓ Wall-clock time          | ~11h         | ~60h   | ~8h         | **~4.5h**      | ~6h         | ~8h      |
| **Qwen3 4B**   | ↑ pass@k        | **0.80**         | **0.80**   | **0.80**        | 0.77       | **0.80**        | 0.83 |
|                | ↑ Pass Rate(**)| 0.62         | 0.63   | **0.68**    | 0.60       | **0.68**    | 0.69 |
|                | ↓ # Tokens (1e5)  | 17           | 11     | 15          | **8**          | 12          | 12       |
|                | ↓ Wall-clock time          | ~12h         | ~25h   | ~10h        | **~5h**    | ~8h         | ~9h      |
| **DeepSeek 8B**| ↑ pass@k        | **0.80**         | **0.80**   | **0.80**        | 0.73       | 0.77        | 0.80 |
|                | ↑ Pass Rate(**)| 0.65         | 0.66   | **0.70**    | 0.59       | 0.68        | 0.71 |
|                | ↓ # Tokens (1e5)  | 22           | 13     | 19          | **8**      | 13          | 12       |
|                | ↓ Wall-clock time          | ~27h         | ~40h   | ~24h        | **~10h**   | ~15h        | ~16h     |
| **GPT-o3s-20B**| ↑ pass@k        | 0.90         | 0.90   | 0.93        | 0.90       | **0.95**    | 0.97 |
|                | ↑ Pass Rate(**)| 0.67         | **0.69**| 0.68       | 0.66       | 0.68        | 0.68     |
|                | ↓ # Tokens (1e5)  | 10           | 7      | 7           | **5**      | **5**       | 5    |
|                | ↓ Wall-clock time          | ~29h         | ~30h   | ~20h        | **~13.5h** | **~13.5h**  | ~17h     |

(*) Unfair advantage having access to the target labels, consider EAGer-adapt for a better comparison. In this table we **bold** best scores excluding EAGer for this reason.

(**) Pass Rate for DeepConf considers all the sequences, those that reached a final answer and those that were stopped before reaching the end.


[1] Deep Think with Confidence, 2025 | online, M = 32 + warmup

[2] Escape Sky-high Cost: Early-stopping Self-Consistency for Multi-step Reasoning, ICLR 2024 | N_windows = 8, M = 32

---

### Author Response · Authors · 2025-12-03
**General response to all reviewers**

**We sincerely thank all reviewers for their thoughtful and constructive feedback.**

Because reviewer responses were frozen due to the OpenReview issue, they could not react to our responses and clarifications; we are sorry they did not have the opportunity to do so and hope this concise recap helps the AC in the evaluation process.

**A summary of the strengths** from all reviewers:
- Clear motivation and simple, intuitive entropy-based method (*Rbnr, veE8, XMDw*).
- Training-free, scalable, easy to deploy, and model-agnostic (*Rbnr, veE8*).
- Strong empirical gains in both accuracy and efficiency across benchmarks (*Rbnr, veE8, F2Hp*).
- Extensive experiments demonstrating adaptive compute allocation and robustness (*F2Hp, XMDw*).

**Our responses to main weaknesses** (all already incorporated into the updated paper):
- Threshold tuning is minimal because the optimal value is stable across models/tasks (*Rbnr*).
- Lack of test-time labels is addressed through EAGer-adapt, which matches efficiency and improves accuracy without oracle signals (*Rbnr*).
- Added strong baselines from related and concurrent work (ESC, DeepConf) and showed EAGer-adapt consistently outperforms them (*veE8, F2Hp*, see general table below).
- Provided expanded efficiency analysis (wall-clock, TTFT, KV-cache behavior) and clarified negligible per-token overhead (*veE8, F2Hp*).
- Improved figure/table clarity, marked EAGer as oracle, and aligned datasets/formatting (*XMDw*).

We again thank the reviewers and the AC for their work. We believe these additions and clarifications directly address the raised concerns and further reinforce the soundness, robustness, and contribution of our approach.

---

### Meta-Review · Area_Chair_QRmV · 2025-12-16

**Summary:**

This paper proposes EAGer, a training-free generation method for test-time scaling. It branches multiple sequences only at high-entropy points ("branch-and-reuse") based on token-level uncertainty and reallocates surplus budget to unresolved prompts (if available).In the rebuttal, the authors addressed major concerns by adding comparisons with relevant baselines (DeepConf/ESC) and efficiency metrics (including at least wall-clock time). They also clarified the positioning of the oracle setting (using ground truth labels) and improved the consistency of tables and figures. While multiple weaknesses have been significantly mitigated, I recommend a Reject in this meta-review for the following reasons: (i) Uncertainty remains regarding design choices (such as the threshold $\theta$) and generality (extrapolation to tasks outside of reasoning). (ii) Particularly regarding "presentation that conveys ideas without misunderstanding," although improvements are visible, I cannot be certain that the reviewers would be fully satisfied.

Based on the foregoing discussion and the reviewers’ scores, and acknowledging how competitive the conference is, I unfortunately recommend rejection.

**Reviewer Concerns:**

* **Comparison with Relevant Baselines (DeepConf/ESC, etc.)** (veE8, F2Hp):
    The positioning of the method has improved with additional comparisons (at least a comparison table on AIME 2025). However, comparisons with DeepConf/ESC are centered mainly on AIME 2025, and direct comparisons under identical conditions across benchmarks remain limited.

* **Depth of Efficiency Evaluation (beyond token count)** (veE8, F2Hp):
    Wall-clock metrics were added, and implementation issues such as KV-cache and TTFT were supplemented in the discussion. Nevertheless, comprehensive verification, including peak memory usage, remains a topic for future work.

* **Clarification of Oracle (Ground Truth Usage) Settings** (XMDw, Rbnr):
    The distinction between EAGer (oracle) and EAGer-adapt (non-oracle/practical assumption) was emphasized, reducing confusion. However, whether readers can immediately grasp "what is possible in actual operation" depends heavily on the presentation quality of the tables and figures.

* **Clarity of Presentation (Consistency of Figures/Tables, Figure 1, etc.)** (XMDw):
    Although figures were reportedly improved and tables made consistent, given the low "Presentation" rating in the initial review, uncertainty remains regarding the clarity of the final manuscript.

* **Hyperparameter Dependence & Generality ($\theta$, $M$, temperature, etc.)** (Rbnr, veE8, XMDw):
    Concerns were mitigated by additional ablations and the claim that "$\theta \approx 2.5$ is generally good." Ultimately, however, the burden of tuning and the prospect of extrapolation to non-reasoning tasks (including theoretical backing) remain open questions.

**Reviewer Scores:**

* **Reviewer Rbnr**: Initial **6** (conf=4).
    While practical aspects improved with the introduction of EAGer-adapt and claims regarding the stability of $\theta$, generality remains an open issue. Overall, likely maintaining **6 → 6** (Moderate confidence).

* **Reviewer F2Hp**: Initial **4** (conf=3).
    Since the requested additional baselines and efficiency analyses were added (at least focusing on AIME 2025) and formatting was corrected, a shift from **4 → 6** is quite possible (Moderate-High confidence).

* **Reviewer veE8**: Initial **4** (conf=3).
    Key requests, such as equal-budget comparison with DeepConf and sensitivity analysis, were reflected. The possibility of **4 → 6** is high (Moderate-High confidence).

* **Reviewer XMDw**: Initial **4** (conf=3).
    Although corrections were made regarding the clarity of Fig. 1, oracle notation, and dataset consistency, given the initial strict stance on Presentation, it is uncertain if they are fully convinced. Maintaining **4 → 4** (Moderate confidence).

**Predicted Average:** (6 + 6 + 6 + 4) / 4 = 5.5.

---

### Decision · Program_Chairs · 2026-01-26

Reject